# LANGUAGE MODELS SCALE RELIABLY WITH OVER-TRAINING AND ON DOWNSTREAM TASKS

**Samir Yitzhak Gadre**[1,2], **Georgios Smyrnis**[3], **Vaishaal Shankar**[4], **Suchin Gururangan**[5],
**Mitchell Wortsman**[5], **Rulin Shao**[5], **Jean Mercat**[2], **Alex Fang**[5], **Jeffrey Li**[5], **Sedrick Keh**[2], **Rui Xin**[5], **Marianna Nezhurina**[6,7], **Igor Vasiljevic**[2], **Jenia Jitsev**[6,7], **Luca Soldaini**[8], **Alexandros G. Dimakis**[9,10], **Gabriel Ilharco**[5], **Pang Wei Koh**[5,8], **Shuran Song**[11], **Thomas Kollar**[2]
**Yair Carmon**[12*], **Achal Dave**[2*], **Reinhard Heckel**[13*], **Niklas Muennighoff**[14*], **Ludwig Schmidt**[5*]

## ABSTRACT

Scaling laws are useful guides for derisking expensive training runs, as they predict performance of large models using cheaper, small-scale experiments. However, there remain gaps between current scaling studies and how language models are ultimately trained and evaluated. For instance, scaling is usually studied in the compute-optimal training regime (i.e., "Chinchilla optimal" regime). In contrast, models are often over-trained to reduce inference costs. Moreover, scaling laws mostly predict loss on next-token prediction, but models are usually compared on downstream task performance. To address both shortcomings, we create a testbed of 104 models with 0.011B to 6.9B parameters trained with various numbers of tokens on three data distributions. First, we fit scaling laws that extrapolate in both the amount of over-training and the number of model parameters. This enables us to predict the validation loss of a 1.4B parameter, 900B token run (i.e., $32\times$ over-trained) and a 6.9B parameter, 138B token run (i.e., a compute-optimal run)—each from experiments that take $300\times$ less compute. Second, we relate the perplexity of a language model to its downstream task performance by proposing a power law. We use this law to predict top-1 error averaged over downstream tasks for the two aforementioned models, using experiments that take $20\times$ less compute. To facilitate further research on reliable scaling, we provide all results of our experiments. Our experiments are available at https://github.com/mlfoundations/scaling.

## 1 INTRODUCTION

Training large language models is expensive. Furthermore, training high-quality models requires a complex recipe of algorithmic techniques and training data. To reduce the cost of finding successful training recipes, researchers first evaluate ideas with small experiments and then extrapolate their efficacy to larger model and data regimes via scaling laws. With reliable extrapolation, it is possible to quickly iterate at small scale and still pick the method that will perform best for the final large training run. Indeed, this workflow has become commonplace for training state-of-the-art language models like Chinchilla 70B (Hoffmann et al., 2022), PaLM 540B (Chowdhery et al., 2022), GPT-4 (OpenAI, 2023), and many others.

Despite their importance for model development, published scaling laws differ from the goals of training state-of-the-art models in important ways. For instance, scaling studies usually focus on the compute-optimal training regime ("Chinchilla optimality" (Hoffmann et al., 2022)), where model and dataset size are set to yield minimum loss for a given compute budget. However, this setting ignores inference costs. As larger models are more expensive at inference, it is now common practice to over-train smaller models (Touvron et al., 2023a). Another potential mismatch is that most scaling laws quantify model performance by perplexity in next-token prediction instead of accuracy on

---

*Equal advising, ordered alphabetically. [1]Columbia University, [2]Toyota Research Institute, [3]UT Austin, [4]Apple, [5]University of Washington, [6]Juelich Supercomputing Center, Research Center Juelich, [7]LAION, [8]Allen Institute for AI, [9]UC Berkeley, [10]Bespoke Labs, [11]Stanford University, [12]Tel Aviv University, [13]TU Munich, [14]Contextual AI

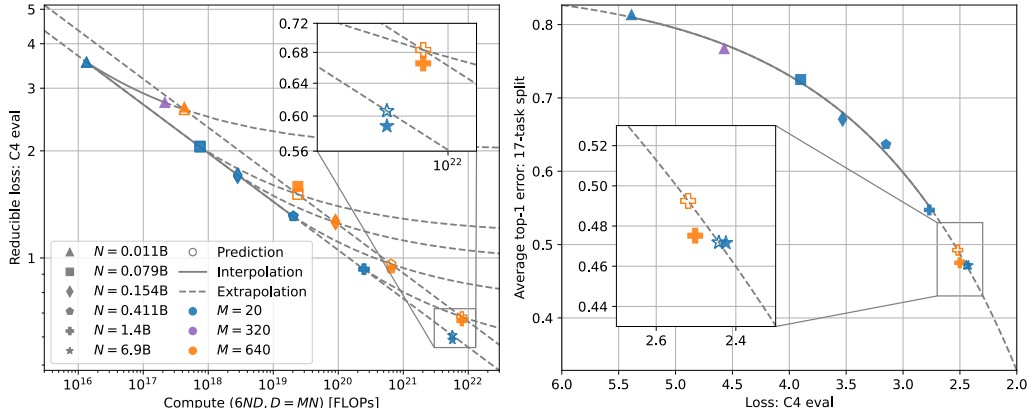

Figure 1: **Reliable scaling with over-training and on downstream error prediction.** *(left)* We fit a scaling law for model validation loss, parameterized by (i) a token multiplier $M = D/N$, which is the ratio of training tokens $D$ to parameters $N$ and (ii) the compute $C$ in FLOPs used to train a model, approximated by $C = 6ND$. Larger values of $M$ specify more over-training. We are able to extrapolate, in both $N$ and $M$, the validation performance of models requiring more than $300\times$ the training compute used to construct the scaling law. *(right)* We also fit a scaling law to predict average downstream top-1 error as a function of validation loss. We find that fitting scaling laws for downstream error benefits from using more expensive models when compared to fitting for loss prediction. We predict the average error over 17 downstream tasks for models trained with over $20\times$ the compute. For this figure, we train all models on RedPajama (Together Computer, 2023).

widely used benchmark datasets. However, practitioners usually turn to benchmark performance, not loss, to compare models.

In this paper, we conduct an extensive set of experiments to address both scaling in the over-trained regime and benchmark performance prediction.

Motivated by the practice of training beyond compute-optimality, we first investigate whether scaling follows reliable trends in the over-trained regime. We notice, as implied by Hoffmann et al. (2022), for a set of models of different sizes trained with a constant ratio of tokens to parameters, models' reducible loss $L'$ (Hestness et al., 2017; Hoffmann et al., 2022) follows a power law ($L' = \lambda \cdot C^{-\eta}$) in the amount of training compute $C$. We find that as one increases the ratio of tokens to parameters, corresponding to more over-training, the scaling exponent $\eta$ remains about the same, while the scalar $\lambda$ changes. We explain our observations by reparameterizing existing scaling laws in relation to the amount of over-training.

To establish empirically that scaling *extrapolates* in the over-trained regime, we further experiment with a testbed of 104 models, trained from scratch on three different datasets: C4 (Raffel et al., 2019; Dodge et al., 2021), RedPajama (Together Computer, 2023), and RefinedWeb (Penedo et al., 2023). We find that scaling laws fit to small models can accurately predict the performance of larger models that undergo more over-training. Figure 1 *(left)* illustrates our main over-training result, where we invest $2.4e19$ FLOPs to extrapolate the C4 validation performance of a 1.4B parameter model trained on 900B tokens, which requires $300\times$ more compute to train.

In addition to over-training, we also investigate if scaling laws can predict the performance of a model on downstream tasks. We establish a power law relationship between language modeling perplexity and the average top-1 error on a suite of downstream tasks. While it can be difficult to predict the error on individual tasks, we find it possible to predict aggregate performance from a model's perplexity among models trained on the same training data. Figure 1 *(right)* presents our main downstream error prediction result, where we invest $2.7e20$ FLOPs to predict the average top-1 error over a set of downstream tasks to within 1 percentage point for a 6.9B compute-optimal model, which requires $20\times$ more compute to train.

Our results suggest that the proposed scaling laws are promising to derisk (i) the effects of over-training models and (ii) the downstream performance of scaling up training recipes. To facilitate further research on reliable scaling, we will provide all results of our experiments.

## 2 DEVELOPING SCALING LAWS FOR OVER-TRAINING AND DOWNSTREAM TASKS

In this section, we develop scaling laws to predict over-trained and downstream performance. First, we provide key definitions (Section 2.1). We next present a scaling law for over-training drawing on empirical observation and prior work (Section 2.2). To connect loss scaling and downstream error prediction, we observe that average top-1 error decreases exponentially as a function of validation loss, which we formalize as a novel scaling law (Section 2.3). In later sections, we build an experimental setup (Section 3) to quantify the extent to which our scaling laws extrapolate reliably (Section 4).

### 2.1 PRELIMINARIES

**Scaling laws for loss.** Typically, scaling laws predict model loss $L$ as a function of the compute $C$ in FLOPs used for training. If one increases the number of parameters $N$ in a model or the number of tokens $D$ that a model is trained on, compute requirements naturally increase. Hence, we assume $C$ is a function of $N, D$. Following Kaplan et al. (2020), we use the approximation $C = 6ND$, which Hoffmann et al. (2022) independently verify. We consider,

$$L(C) = E + L'(C), \tag{1}$$

where $E$ is an *irreducible loss* and $L'$ is the *reducible loss*. $E$ captures the Bayes error or minimum possible loss achievable on the validation domain. The $L'(C)$ term captures what can possibly be learned about the validation domain by training on a source domain. $L'(C)$ should approach zero with increased training data and model capacity. $L'(C)$ is often assumed to follow a power law: $L'(C) = \lambda \cdot C^{-\eta}$ (i.a., Hestness et al. (2017); OpenAI (2023)). It is also often helpful to consider a power law in a $\log$-$\log$ plot, where it appears as a line with slope $-\eta$ and $y$-intercept $\log(\lambda)$.

**Token multipliers.** We define a token multiplier $M = D/N$ as the ratio of training tokens to model parameters for notational convenience. $M$ allows us to consider fixed relationships between $D$ and $N$ even as a model gets bigger (i.e., as $N$ becomes larger).

**Compute-optimal training.** Hoffmann et al. (2022) establish compute-optimal training, where, for any compute budget $H$, the allocation of parameters and tokens is given by,

$$\arg \min_{N,D} L(N, D) \text{ s.t. } C(N, D) = H. \tag{2}$$

To solve for the optimal $N^*, D^*$, one can sweep $N, D$ for each compute budget, retaining the best configurations. Hoffmann et al. (2022) find that as the compute budget increases, $N^*$ and $D^*$ scale roughly evenly. Assuming equal scaling, there is a fixed compute-optimal token multiplier $M^* = D^*/N^*$ per training distribution.

**Over-training.** We define over-training as the practice of allocating compute sub-optimally, so smaller models train on a disproportionately large number of tokens (i.e., $M > M^*$). While loss should be higher than in the compute-optimal allocation for a given training budget, the resulting models have fewer parameters and thus incur less inference cost.

### 2.2 SCALING LAWS FOR OVER-TRAINING

To propose a scaling law for over-trained models, we first turn to empirical observation. We train four model configurations with parameter counts between 0.011B and 0.411B for token multipliers $M$ between 20 and 640, where $M = 20$ points lie roughly on the compute-optimal frontier, and larger $M$ corresponds to more over-training. We defer experimental details to Section 3 to focus on our observations first. In Figure 2, we show loss against compute in a $\log$-$\log$ plot for the models trained on three datasets and evaluated on the C4 eval set. We notice parallel lines when fitting power laws to

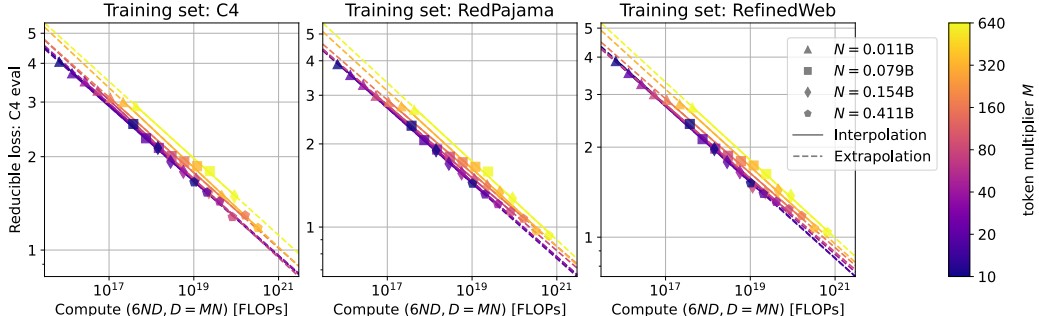

Figure 2: **Scaling in the over-trained regime follows consistent power law exponents.** We notice parallel lines in the $\log$-$\log$ plots of reducible loss vs. training compute for a range of token multipliers $M$, which give the ratio of training tokens to model parameters. Larger $M$ corresponds to more over-training. For a power law giving reducible loss as a function of compute: $L'(C) = \lambda \cdot C^{-\eta}$, the exponent $\eta$ remains relatively constant resulting in lines with approximately fixed slope (Figure 17). The scalar $\lambda$ that determines the $y$-intercept, however, shifts with different token multipliers. This suggests $\lambda$ is a function of the token multiplier, while $\eta$ is not.

the reducible loss, which suggests a near-constant scaling exponent even with increased over-training. This indicates that scaling behavior should be describable in the amount of over-training.

In search of an analytic expression for the observations in Figure 2, we consider existing scaling literature. A common functional form for the risk of a model, as proposed in prior work (Rosenfeld et al., 2020; Hoffmann et al., 2022) is,

$$L(N, D) = E + AN^{-\alpha} + BD^{-\beta}. \tag{3}$$

Recall from Section 2.1, $N$ is the number of parameters and $D$ the number of training tokens. The constants $E, A, \alpha, B, \beta$ are fit from data. By fitting this parametric form, Hoffmann et al. (2022) find that scaling exponents $\alpha$ and $\beta$ are roughly equal, suggesting that one should scale $N$ and $D$ equally as compute increases. Hence, we assume $\alpha = \beta$. With this assumption, we reparameterize Equation (3) in terms of compute $C = 6ND$ and a token multiplier $M = D/N$. We get,

$$L(C, M) = E + \left(aM^\eta + bM^{-\eta}\right) C^{-\eta}, \tag{4}$$

where $\eta = \alpha/2$, $a = A(1/6)^{-\eta}$, $b = B(1/6)^{-\eta}$ gives the relation to Equation (3). For a complete derivation, see Appendix A.

Equation (4) has the following interpretation: (i) The scaling exponent $\eta$ is not dependent on $M$. Thus, we always expect lines with the same slope in the $\log$-$\log$ plot—as in Figure 2. (ii) The term $aM^\eta + bM^{-\eta}$ determines the offsets between curves with different token multipliers. Hence, we expect non-overlapping, parallel lines in the $\log$-$\log$ plot for the range of $M$ we consider—also consistent with Figure 2.

Recall that we make the assumption $\alpha = \beta$, which implies equal scaling of parameters and tokens as more compute is available. However, as explained in Appendix A, even if $\alpha \neq \beta$, we get a parameterization that implies the power-law exponent remains constant with over-training.

## 2.3 SCALING LAWS FOR DOWNSTREAM ERROR

Scaling is typically studied in the context of loss (Kaplan et al., 2020; Hoffmann et al., 2022; Muennighoff et al., 2023b), which Schaeffer et al. (2023) note is smoother than metrics like accuracy. However, practitioners often use downstream benchmark accuracy as a proxy for model quality and not loss on perplexity evaluation sets. To better connect scaling laws and over-training to task prediction, we revisit the suite of models plotted in Figure 2. In Figure 3, we plot average downstream top-1 errors over evaluations sourced from LLM-Foundry (MosaicML, 2023) against the C4 eval loss. We defer details of the setup to Section 3 to focus here on a key observation: average error appears to follow exponential decay as loss decreases.

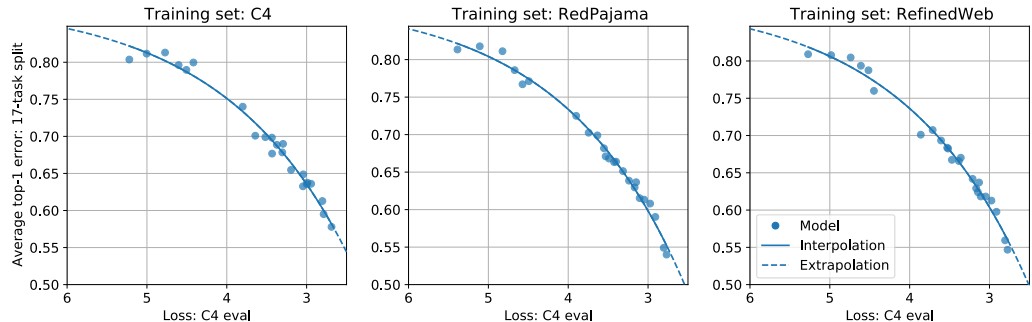

Figure 3: **Average top-1 error scales as a function of loss.** We plot models trained on three datasets and notice an exponential decay of average top-1 error as C4 eval loss, on the x-axis, decreases. We consider on the y-axes average error on 17 evaluations where performance is at least 10 points above random chance for at least one 0.154B scale model. These observations suggest that average top-1 error should be predictable with reliable loss estimates.

Based on the exponential decay we observe in Figure 3, we propose the following relationship between downstream average top-1 error $\mathsf{Err}$ and loss $L$,

$$\mathsf{Err}(L) = \epsilon - k \cdot \exp\left(-\gamma L\right), \tag{5}$$

where $\epsilon, k, \gamma$ are fit from data. Equation (5) also has an interpretation in terms of model perplexity $\mathsf{PP}(L) = \exp\left(L\right)$,

$$\mathsf{Err}(\mathsf{PP}) = \epsilon - k \cdot \mathsf{PP}^{-\gamma}. \tag{6}$$

Namely, $\mathsf{Err}$ follows a power law in $\mathsf{PP}$ that is bounded from above by $\epsilon$ signifying arbitrarily high error and from below by $\epsilon - k \cdot \exp(-\gamma E)$, where $E$ is the Bayes error from Equation (4).

Equation (5) in conjunction with Equation (4) suggests a three-step method to predict $\mathsf{Err}$ as a function of compute and the amount of over-training. For choices of training and validation distributions, (i) fit a scaling law to Equation (4) using triplets of compute $C$, token multiplier $M$, and measured loss $L$ on a validation set to yield $(C, M) \mapsto L$. (ii) Fit a scaling law to Equation (5) using pairs of loss $L$ and downstream error $\mathsf{Err}$ for models to get $L \mapsto \mathsf{Err}$. (iii) Chain predictions to get $(C, M) \mapsto \mathsf{Err}$.

## 3 CONSTRUCTING A SCALING TESTBED

In this section, we discuss our experimental setup to test the predictions suggested by Equations (4) and (5). We first present our general language modeling setup (Section 3.1). Next, we discuss our strategy for determining model configurations for our scaling investigation (Section 3.2) and fitting scaling laws (Section 3.3). We then present metrics to validate how well scaling laws predict loss and downstream performance (Section 3.4).

### 3.1 TRAINING SETUP

We train transformers (Vaswani et al., 2017) for next token prediction, based on architectures like GPT-2 (Radford et al., 2019) and LLaMA (Touvron et al., 2023a). We employ GPT-NeoX (Black et al., 2022) as a standardized tokenizer for all data. See Appendix B for architecture, optimization, and hyperparameter details.

### 3.2 MODEL CONFIGURATIONS

To get final configurations for the 0.011B to 0.411B parameter models plotted in Figures 2 and 3, we first conduct a wide grid search over a total of 435 models, trained from scratch, from 0.01B to 0.5B parameters (Figure 4 *(left)*). We train on the original OpenLM data mix (Gururangan et al., 2023), which largely consists of RedPajama (Together Computer, 2023) and The Pile (Gao et al., 2020). While we eventually plan to over-train models, at this step we search for *base configurations* near

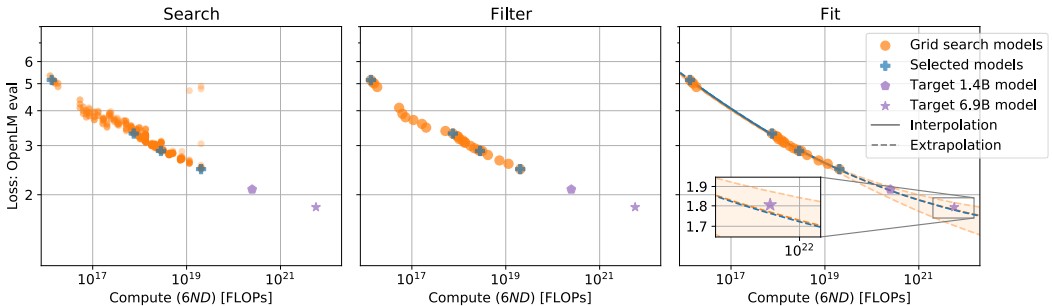

Figure 4: **Search, filter, fit: A recipe for selecting configurations for scaling.** *(left)* To generate the final configurations presented in Table 3, we run a 435 model grid search over model width, hidden dimension, number of attention heads, batch size, and warmup steps. All models are trained near compute-optimally. *(center)* We plot the efficient frontier of models, which appear to follow a trend, excluding models from $5.2 \times 10^{16}$ to $5.2 \times 10^{17}$, which fall below the trend. *(right)* We fit a power law with irreducible error to the remaining configurations, picking four configurations that closely track the full model suite ("Selected models"). These models extrapolate the performance of 1.4B, 6.9B target models. Shaded regions represent bootstrap 95% confidence intervals.

compute-optimality. We train on 20 tokens per parameter ($M = 20$), which, in early experiments, gives models near the compute-optimal frontier. This is similar to findings in Hoffmann et al. (2022)'s Table 3, which suggests that $M = 20$ is near-optimal for the Chinchilla experimental setup.

To find maximally performant small-scale models on validation data, we tune model width, number of layers, number of attention heads, warmup steps, and batch size. Our validation set, OpenLM eval, contains tokens from recent arXiv papers, the OpenLM codebase itself, and news articles. We find in early experiments that qk-LayerNorm makes models less sensitive to learning rate, which is a phenomenon Wortsman et al. (2023) report in their Figure 1. Hence, we fix the learning rate ($3e$-3) for our sweeps. We also perform smaller grid searches over 1.4B and 6.9B parameter model configurations at $M = 20$, retaining the best configurations.

At this point, we have many models, several of which give poor performance; following prior work (Kaplan et al., 2020; Hoffmann et al., 2022), we want to keep only models that give best performance. Hence, in Figure 4 *(center)*, we filter out models that do not lie on the Pareto frontier. While there appears to be a general trend, configurations between $5.2 \times 10^{16}$ and $5.2 \times 10^{17}$ FLOPs lie below the frontier established by other models. We hypothesize these models over-perform as they are trained for more optimization steps than their neighbors based on our power-of-two batch sizes. We provide support for this hypothesis in Appendix E, but opt to remove these models from our investigation.

To ensure tractable compute requirements for our scaling experiments, we require a subset of models that follows the trend of the entire Pareto frontier. In Figure 4 *(right)*, we fit trends to the Pareto models and to a subset of four models. We notice that the trends closely predict both the performance of the 1.4B and 6.9B models, suggesting that our small-scale configurations reliably extrapolate in the compute-optimal setting.

Moving forward, we do not tune hyperparameters for other token multipliers (i.e., $M \neq 20$), on other training or evaluation distributions, or on validation sets for downstream tasks. For more details including specific hyperparameters, see Appendix C.

To create our scaling testbed, we start with the four small-scale, base configurations from our grid search: $N \in \{0.011\text{B}, 0.079\text{B}, 0.154\text{B}, 0.411\text{B}\}$. To ensure our conclusions are not particular to a single training distribution, we train models on each of C4 (Raffel et al., 2019; Dodge et al., 2021), RedPajama (Together Computer, 2023), and RefinedWeb (Penedo et al., 2023), which have 138B, 1.15T, and 600B tokens, respectively, for different token multipliers $M \in \{5, 10, 20, 40, 80, 160, 320, 640\}$. We omit runs that require more tokens than are present in a dataset (i.e., $N = 0.411\text{B}, M = 640$ for C4). We additionally train $N = 1.4\text{B}$ models at $M = 20$ and at the largest token multiplier possible without repeating tokens (i.e., 80 for C4, 640 for

Table 1: **Default number of parameters $N$ and token multiplier $M$ to fit our scaling laws.** We invest $\sim$100 A100 hours to fit Equation (4) and $\sim$1,000 A100 hours to fit Equation (5).

| $N$ | $M$ | Used to fit Equation (4) | Used to fit Equation (5) |
|---|---|---|---|
| 0.011B | 20 | ✓ | ✓ |
| 0.079B | 20 | ✓ | ✓ |
| 0.154B | 20 | ✓ | ✓ |
| 0.411B | 20 | ✓ | ✓ |
| 0.011B | 320 | ✓ | ✓ |
| 1.4B | 20 | ✗ | ✓ |
| Total compute $C$ [FLOPs] | | 2.4$e$19 | 2.7$e$20 |

RedPajama, and 320 for RefinedWeb). We train $N = 6.9$B, $M = 20$ models on each dataset given the relevance of 7B parameter models (Touvron et al., 2023a; Jiang et al., 2023). In total this results in a testbed of 104 models.

### 3.3 Fitting scaling laws

We fit Equation (4) to approximate $E, a, b, \eta$ using curve-fitting in SciPy (Virtanen et al., 2020) (i.e., Levenberg-Marquardt to minimize non-linear least squares). We repeat this process to fit Equation (5) to approximate $\epsilon, k, \gamma$. We invest $\sim$100 A100 hours to train the models required to fit a scaling law for loss and $\sim$1,000 A100 hours for a corresponding law for downstream error. Unless otherwise specified, we fit to the $N, M$ pairs in Table 1, which are a subset of our full testbed. Our configurations allow us to test for extrapolation to the $N = 1.4$B, $M = 640$ (900B token) and the $N = 6.9$B, $M = 20$ (138B token) regimes.

### 3.4 Evaluation setup

**Evaluation datasets.** Unless otherwise stated, our default validation loss dataset is C4 eval. For downstream tasks, we adopt a subset from 46 tasks from LLM-foundry (MosaicML, 2023), which includes standard tasks with both zero-shot and few-shot evaluations. Specifically, we consider a 17-task subset where, for each evaluation, at least one 0.154B scale model—trained with as many as 99B tokens—gets 10 percentage points above chance accuracy: ARC-Easy (Clark et al., 2018), BIG-bench: CS algorithms (bench authors, 2023), BIG-bench: Dyck languages (bench authors, 2023), BIG-bench: Novel Concepts (bench authors, 2023), BIG-bench: Operators (bench authors, 2023), BIG-bench: QA WikiData (bench authors, 2023), BoolQ (Clark et al., 2019), Commonsense QA (Talmor et al., 2019), COPA (Roemmele et al., 2011), CoQA (Reddy et al., 2019), HellaSwag (zero-shot) (Zellers et al., 2019), HellaSwag (10-shot) (Zellers et al., 2019), LAMBADA (Paperno et al., 2016), PIQA (Bisk et al., 2020), PubMed QA Labeled (Jin et al., 2019), SQuAD (Rajpurkar et al., 2016), and WinoGrand (Levesque et al., 2012). For more details on evaluation datasets see Appendix D. We focus on this subset to ensure we are measuring signal, not noise. Including downstream tasks like MMLU (Hendrycks et al., 2021), where performance is close to random chance, however, does not invalidate our results as we show in our evaluation set ablations (Appendix E).

**Metrics.** We consider three main metrics: *Validation loss*, which is the cross entropy between a model's output and the one-hot ground truth token, averaged over all tokens in a sequence and over all sequences in a dataset. *Average top-1 error*, which is a uniform average over the 17 downstream evaluations, as mentioned in the above paragraph. To measure how good a prediction $\zeta(C, M)$ is, we measure *Relative prediction error*: $|\zeta(C, M) - \zeta_{GT}|/\zeta_{GT}$, where $\zeta$ is the predicted loss $L$ or the average top-1 error Err. $\zeta_{GT}$ is the ground truth measurement to predict.

## 4 Results: Reliable extrapolation

In this Section, we quantify the extent to which the scaling laws developed in Section 2 extrapolate larger model performance using the scaling testbed from Section 3. By default, we fit Equations (4)

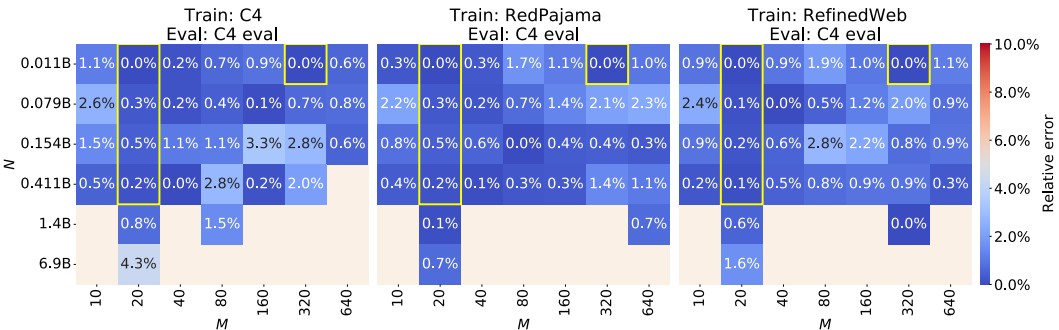

Figure 5: **Relative error on C4 eval for different training distributions.** Boxes highlighted in yellow correspond to pairs—number of parameters $N$, token multiplier $M$—used to fit Equation (4). Larger values of $M$ correspond to more over-training. The prediction error is low in both interpolation and extrapolation ranges. Below $N = 1.4$B, empty squares correspond to runs that were not possible due to the limited dataset size for single epoch training. At $N = 1.4$B we run at $M = 20$ and at the largest possible multiplier. At $N = 6.9$B, we run at $M = 20$.

and (5) to the configurations in Table 1, use C4 eval for loss, and the 17-task split from Section 3.4 for average top-1 error.

**Over-trained performance is predictable.** We highlight our main over-training results in Figure 1 *(left)*. Namely, we are able to extrapolate both in the number of parameters $N$ and the token multiplier $M$ to closely predict the C4 eval performance of a 1.4B parameter model trained on 900B RedPajama tokens ($N = 1.4$B, $M = 640$). Our prediction, which takes $300\times$ less compute to construct than the final 1.4B run, is accurate to within 0.7% relative error. Additionally, for the $N = 6.9$B, $M = 20$ run, near compute-optimal, the relative error is also 0.7%.

These results support several key takeaways. (i) Scaling can be predictable even when one increases both the model size and the amount of over-training compared to the training runs used to fit a scaling law. (ii) The form presented in Equation (4) is useful in practice for predicting over-trained scaling behavior. (iii) Fitting to Equation (4) gives good prediction accuracy near compute-optimal. More specifically, predictions are accurate both for the 1.4B over-trained model and the 6.7B compute-optimal model using a single scaling fit.

While Figure 1 explores a specific case of making predictions in the over-trained regime, we aim to understand the error profile of our predictions across training datasets, token multipliers, and number of parameters. Hence, Figure 5 shows the relative error between ground truth loss and predicted loss on C4 eval for models in our testbed. We notice uniformly low prediction error suggesting that predictions are accurate in many settings.

**Average top-1 error is predictable.** Figure 1 *(right)* presents our main result in estimating scaling laws for downstream error. Concretely, we use the models indicated in Table 1 to fit Equations (4) and (5), chaining the scaling fits to predict the average top-1 error as a function of training compute $C$ and the token multiplier $M$. Our fits allow us to predict, using $20\times$ less compute, the downstream performance of a 6.9B model trained on 138B RedPajama tokens to within 0.05% relative error and a 1.4B model trained on RedPajama 900B tokens to within 3.6% relative error.

Table 2 additionally shows the relative error of our downstream performance predictions for models trained on C4, RedPajama, and RefinedWeb, indicating that our scaling law functional forms are applicable on many training datasets. We note that while average accuracy is predictable, *individual* downstream task predictions are significantly more noisy. We report relative error for more model predictions in Figures 11 and 12. We also find that if we remove the 1.4B model for the Equation (5) fit, relative error jumps, for instance, from 0.05% to 10.64% on the 17-task split for the 6.9B, 138B token RedPajama prediction. This highlights the importance of investing more compute when constructing scaling laws for downstream task prediction compared to loss prediction.

Table 2: **Downstream relative prediction error at 6.9B parameters and 138B tokens.** While predicting accuracy on individual zero-shot downstream evaluations can be challenging ("Individual"), predicting *averages* across downstream datasets is accurate ("Avg.").

| Train set | Individual top-1 error | | | | Avg. top-1 error |
|---|---|---|---|---|---|
| | ARC-E | LAMBADA | OpenBook QA | HellaSwag | 17-task split |
| C4 | 28.96% | 15.01% | 16.80% | 79.58% | 0.14% |
| RedPajama | 5.21% | 14.39% | 8.44% | 25.73% | 0.05% |
| RefinedWeb | 26.06% | 16.55% | 1.92% | 81.96% | 2.94% |

**Under-training, out-of-distribution scaling, compute-reliability trade-offs.** In addition to our main results presented above, we include additional results in Appendix E, which we summarize here. First, we notice that when token multipliers become too small (i.e., $M = 5$) scaling becomes unreliable and lies off the trend. Additionally, multipliers other than 20, such as 10, 40, and 80, garner points that are roughly on the compute optimal frontier (Figure 9). This observation suggests that the compute-optimal multiplier may lie in a range rather than take a single value. To probe the limits of reliable scaling, we attempt to break our scaling laws in out-of-distribution settings. We find that models trained on C4—English filtered—and evaluated on next token prediction on code domains have a high relative error in many cases. Perhaps surprisingly, evaluating the same models on German next token prediction gives reliable loss scaling (Figure 10). We additionally examine the compute necessary to create accurate scaling laws, finding that scaling laws can be constructed more cheaply for loss prediction than for downstream error prediction (Figures 15 and 16).

## 5 RELATED WORK

We review the most closely related work in this section. For additional related work, see Appendix F.

**Scaling laws.** Early works on scaling artificial neural networks observe predictable power-law scaling in the training set size and number of model parameters (Hestness et al., 2017; 2019; Rosenfeld et al., 2020). Alabdulmohsin et al. (2022) stress the importance of looking at the extrapolation regime of a scaling law. Yang et al. (2021) prescribe architectural and hyperparameter changes when scaling model width to realize performant models; Yang et al. (2024) make analogous recommendations when scaling model depth. Bi et al. (2024) propose hyperparameter aware scaling laws. Unlike the aforementioned work, our investigation focuses on over-training and predicting downstream accuracy.

Hoffmann et al. (2022) investigate how the number of model parameters $N$ and training tokens $D$ should be chosen to minimize loss $L$ given a compute budget $C$. Hoffmann et al. (2022) find that when scaling up $C$, both $N$ and $D$ should be scaled equally up to a multiplicative constant (i.e., $N \propto C^{\sim 0.5}$ and $D \propto C^{\sim 0.5}$) to realize compute-optimality. Appendix C of the Chinchilla paper additionally suggests that these findings hold across three datasets. However, Hoffmann et al. (2022) do not verify their scaling laws for training beyond compute-optimality, or for downstream error prediction—both of which are central to our work.

Sardana & Frankle (2023) propose modifications to the Chinchilla formulation to incorporate inference costs into the definition of compute-optimality and solve for various fixed inference budgets. Their key finding, which is critical for our work, is that when taking into account a large enough inference budget, it is optimal to train smaller models for longer than the original Chinchilla recommendations. Our work presupposes that over-training can be beneficial. Instead of solving for inference-optimal schemes, we support empirically a predictive theory of scaling in the over-trained regime. Additionally, we provide experiments across many validation and training sets.

For predicting downstream scaling beyond loss, Isik et al. (2024) relate the number of pre-training tokens to downstream cross-entropy and machine translation BLEU score (Papineni et al., 2002) after fine-tuning. In contrast, we take a holistic approach to evaluation by looking at top-1 error over many natural language tasks. Schaeffer et al. (2023) argue that emergent abilities (Wei et al., 2022b) are a product of non-linear metrics and propose smoother alternatives. As a warmup for why non-linear metrics may be hard to predict, Schaeffer et al. (2023) consider predicting an $\ell$ length sequence

exactly: $\mathsf{Err}(N, \ell) \approx 1 - \mathsf{PP}(N)^{-\ell}$, where $N$ is the number of parameters in a model and $\mathsf{PP}$ is its perplexity. This is a special case of our Equations (5) and (6), where the number of training tokens does not appear, $\epsilon = 1, k = 1$, and $\gamma = \ell$. In contrast, we treat $\epsilon, k, \gamma$ as free parameters for a scaling law fit, finding that average error over downstream tasks can make for a predictable metric. Owen (2024) observe the scaling behavior of many open source models on downstream tasks. However, their study does not control for different architectures, training codebases, optimization schemes, and training datasets. We create a standardized, open-source setting, which controls these factors.

**Over-training in popular models.** There has been a rise in over-trained models (Touvron et al., 2023a;b; Llama Team, 2024) and accompanying massive datasets (Together Computer, 2023; Penedo et al., 2023; Soldaini et al., 2024; Albalak et al., 2024). For example, Chinchilla 70B (Hoffmann et al., 2022) is trained with a token multiplier of 20, while Llama-2 7B (Touvron et al., 2023b) uses a token multiplier of 290. In our investigation, we look at token multipliers from 5 to 640 for coverage of popular models. The recent Llama3 8B model is a notable outlier, with token multipliers of ∼1900. However, it is unclear if, at 15T tokens, Llama3 8B was trained in the single epoch regime we considr in this paper. Practically, training a 1.4B parameter model at this multiplier is prohibitive due to 1) compute limitations and 2) the 2.8T training token requirement for a single-epoch run, which is larger than public datasets at the time of our training runs.

## 6 LIMITATIONS, FUTURE WORK, AND CONCLUSION

**Limitations and future work.** We identify limitations, which provide motivation for future work.

- **Hyperparameters.** While our configurations are surprisingly amenable to reliable scaling across many training and testing distributions without further tuning, there is a need to develop scaling laws that do not require extensive hyperparameter sweeps.
- **Scaling up.** Validating the trends in this paper for even larger runs is a valuable direction. Additionally, repeating our setup for models that achieve non-trivial performance on harder evaluations like MMLU is left to future work.
- **Scaling down.** Actualizing predictable scaling with even cheaper runs is important to make this area of research more accessible, especially for downstream error prediction.
- **Failure cases.** While we present a preliminary analysis of when scaling is unreliable, future work should investigate conditions under which scaling breaks down.
- **Post-training.** It is common to employ fine-tuning interventions after pre-training, which we do not consider. Quantifying to what degree over-training the base model provides benefits *after* post-training is an open area of research.
- **Individual downstream task prediction.** Accurate per-task predictions are left to future work.
- **In-the-wild performance.** Downstream task performance is a proxy for the in-the-wild user experience. Analyzing scaling trends in the context of this experience is timely.
- **Dataset curation.** Our work only deals with existing training datasets. Exploring dataset curation for improved model scaling is another promising direction.

**Conclusion.** We show that the loss of over-trained models, trained past compute-optimality, is predictable. Furthermore, we propose and validate a scaling law relating loss to average downstream task performance. We hope our work will inspire others to further examine the relationship between model training and downstream generalization. Our testbed will be made publicly available, and we hope it will make scaling research more accessible to researchers and practitioners alike.

## ACKNOWLEDGEMENTS

SYG is supported by an NSF Graduate Research Fellowship, GS by the Onassis Foundation - Scholarship ID: F ZS 056-1/2022-2023, and MN by the Federal Ministry of Education and Research of Germany under grant no. 01IS22094B WEST-AI. We thank Stability AI and Toyota Research Institute (TRI) for access to compute resources. This research has been supported by NSF Grants AF 1901292, CNS 2148141, Tripods CCF 1934932, IFML CCF 2019844, and research gifts by Western Digital, Amazon, WNCG IAP, UT Austin Machine Learning Lab (MLL), Cisco, and the Stanly P. Finch Centennial Professorship in Engineering. We also thank Kushal Arora, Alper Canberk, Mia Chiquier, Sachit Menon, Mariah Oxley, Chuer Pan, Purva Tendulkar, and Mandi Zhao for valuable feedback.

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
