CONTENTS

# A  Scaling-law derivations

We first show that reparameterizing Equation (3) in terms of the compute $C$ and token multiplier $M$ for $\alpha = \beta$ yields Equation (4). Combining $C = 6ND$ and $M = D/N$ yields $N = \sqrt{C/(6M)}$ and $D = \sqrt{CM/6}$. Inserting these into Equation (3) yields,

$$
\begin{aligned}
L(C, M) &= E + A \left( \frac{C}{6M} \right)^{-\frac{\alpha}{2}} + B \left( \frac{CM}{6} \right)^{-\frac{\alpha}{2}}, \\
&= E + \left( A \left( \frac{1}{6} \right)^{-\frac{\alpha}{2}} M^{\frac{\alpha}{2}} + B \left( \frac{1}{6} \right)^{-\frac{\alpha}{2}} M^{-\frac{\alpha}{2}} \right) C^{-\frac{\alpha}{2}}.
\end{aligned}
$$

This is equal to Equation (4), making the substitutions $\eta = \alpha/2$, $a = A(1/6)^{-\eta}$, $b = B(1/6)^{-\eta}$, as noted in the main body.

**Relation to compute-optimal training.**  Recall that we made the assumption $\alpha = \beta$, which implies equal scaling of parameters and tokens to realize compute-optimal models. While this assumption is empirically justified (Hoffmann et al., 2022), even if $\alpha \neq \beta$, we get a parameterization that implies the power law exponent in Equation (4) remains constant with over-training, while the power law scalar changes.

To find a compute-optimal training setting, Hoffmann et al. (2022) propose to minimize the right-hand side of Equation (3) subject to the compute constraint $C = 6ND$. This yields, $N^* = \gamma^{\frac{1}{\alpha+\beta}} (C/6)^{\frac{\beta}{\alpha+\beta}}$ and $D^* = \gamma^{-\frac{1}{\alpha+\beta}} (C/6)^{\frac{\alpha}{\alpha+\beta}}$, where $\gamma = \frac{\alpha A}{\beta B}$, for notational convenience. The associated risk is,

$$
L(N^*, D^*) = E + \left( A\gamma^{\frac{-\alpha}{\beta+\alpha}} + B\gamma^{\frac{\beta}{\beta+\alpha}} \right) \left( \frac{C}{6} \right)^{-\frac{\alpha\beta}{\alpha+\beta}}.
$$

We now deviate from compute-optimal training by modifying the model size and tokens by multiplication with a constant $\sqrt{m}$, according to

$$
N_m = \frac{1}{\sqrt{m}} N^*, \quad D_m = \sqrt{m} D^*. \tag{7}
$$

This modification keeps the compute constant (i.e., $6N_m D_m = 6N^* D^*$). The risk, then, becomes

$$
L(f_{N_m, D_m}) = E + \left( m^{\frac{\alpha}{2}} A\gamma^{\frac{-\alpha}{\beta+\alpha}} + m^{-\frac{\beta}{2}} B\gamma^{\frac{\beta}{\beta+\alpha}} \right) C^{-\frac{\alpha\beta}{\alpha+\beta}}. \tag{8}
$$

We again expect the same power law exponent and changing power law scalar. Note that $m$ in Equation (8) is similar to $M$ in Equation (4). Specifically, $m$ is a multiple of the Chinchilla-optimal token multiplier $M^* = D^*/N^*$, which is no longer fixed as a compute budget changes for $\alpha \neq \beta$.

Table 3: **Main models and hyperparameters used in our investigation.** Models have number of parameters $N$, with number of layers $n_{\text{layers}}$, number of attention heads $n_{\text{heads}}$, model width $d_{\text{model}}$, and width per attention head $d_{\text{head}}$. Batch sizes are global and in units of sequences. Each sequence has 2,048 tokens. A100 GPU hours are at $M = 20$, which are near compute-optimal runs. For the 1.4B scale, a batch size of 256 performs slightly better than 512.

| $N$ | $n_{\text{layers}}$ | $n_{\text{heads}}$ | $d_{\text{model}}$ | $d_{\text{head}}$ | Warmup | Learning rate | Batch size | $M = 20$ A100 hours |
|---|---|---|---|---|---|---|---|---|
| 0.011B | 8 | 4 | 96 | 24 | 100 | $3e$-3 | 64 | 0.3 |
| 0.079B | 8 | 4 | 512 | 128 | 400 | $3e$-3 | 512 | 5 |
| 0.154B | 24 | 8 | 576 | 72 | 400 | $3e$-3 | 512 | 12 |
| 0.411B | 24 | 8 | 1,024 | 128 | 2,000 | $3e$-3 | 512 | 75 |
| 1.4B | 24 | 16 | 2,048 | 128 | 5,000 | $3e$-3 | 256 | 690 |
| 6.9B | 32 | 32 | 4,096 | 128 | 5,000 | $3e$-4 | 2,048 | 17,000 |

## B  ADDITIONAL TRAINING DETAILS

**Architecture.**  As stated in the main paper, we train transformers (Vaswani et al., 2017), based on auto-regressive, decoder-only, pre-normalization architectures like GPT-2 (Radford et al., 2019) and LLaMA (Touvron et al., 2023a). We adopt OpenLM (Gururangan et al., 2023) for modeling, which utilizes PyTorch (Paszke et al., 2019; Ansel et al., 2024), xformers (Lefaudeux et al., 2022), triton (OpenAI, 2021), FlashAttention (Dao et al., 2022), FSDP (Zhao et al., 2023), and bfloat16 automatic mixed precision. Like LLaMA, we omit bias terms, but replace RMSNorm (Zhang & Sennrich, 2019) with LayerNorm (Ba et al., 2016), which has readily available fused implementations. Following Wortsman et al. (2023), we apply qk-LayerNorm (Dehghani et al., 2023), which adds robustness to otherwise poor hyperparameter choices (e.g., learning rate). We use SwiGLU (Shazeer, 2020) activations and depth-scaled initialization (Zhang et al., 2019). We use a sequence length of 2,048, rotary positional embeddings (Su et al., 2021), and the GPT-NeoX-20B tokenizer (Black et al., 2022), which yields a vocabulary size of 50k. We do not use weight tying (Press & Wolf, 2017; Inan et al., 2017). We sample without replacement during training and employ sequence packing without attention masking. We separate documents in our training corpora with end-of-text tokens.

**Objectives and optimization.**  We train with a standard causal language modeling objective (i.e., next token prediction) with an additive z-loss (Chowdhery et al., 2022) (coefficient $1e$-4), which mitigates output logit norm growth (Merrill et al., 2021) instabilities. We use the AdamW optimizer (Loshchilov & Hutter, 2017) (PyTorch defaults except `beta2 = 0.95`), with independent weight decay (Wortsman et al., 2023) (coefficient $1e$-4). For the learning rate schedule, we use linear warmup and cosine decay. We cool down to a low learning rate ($3e$-5).

## C  ADDITIONAL GRID SEARCH DETAILS

**Final model configurations.**  We present our final hyperparameters in Table 3.

**Grid search configuration selection.**  Recall in Section 3.3, we run a grid search over many configurations. We present the architectures we sweep over in Table 4.

## D  EVALUATION DATASET DETAILS

All 46 downstream evaluations are based on MosaicML's LLM-foundry evaluation suite (MosaicML, 2023). We specifically consider the datasets given in Table 5. Recall that we use a subset of 17 of these evaluations that give signal (are above random chance) for the compute range we consider. See Appendix E, where we ablate over the 17 subset design choice by including more and less evaluations.

## E  ADDITIONAL RESULTS

**Scaling law fits.**  We present specific coefficients for our fits in Table 6.

Table 4: **Topologies for our grid searches.** We consider 130 architectures for our grid search. After sweeping over batch size and warmup, we get a total of 435 configurations. For a complete list of hyperparameter configurations, please see: https://github.com/mlfoundations/scaling

| $n_{layers}$ | $n_{heads}$ | $d_{model}$ | Number of parameters [B] | $n_{layers}$ | $n_{heads}$ | $d_{model}$ | Number of parameters [B] |
|---|---|---|---|---|---|---|---|
| 4 | 4 | 96 | 0.010 | 12 | 4 | 512 | 0.093 |
| 4 | 12 | 96 | 0.010 | 16 | 12 | 488 | 0.100 |
| 12 | 12 | 96 | 0.011 | 8 | 16 | 640 | 0.105 |
| 12 | 4 | 96 | 0.011 | 8 | 4 | 640 | 0.105 |
| 8 | 4 | 96 | 0.011 | 8 | 8 | 640 | 0.105 |
| 16 | 4 | 96 | 0.011 | 12 | 8 | 576 | 0.106 |
| 16 | 12 | 96 | 0.011 | 16 | 16 | 512 | 0.106 |
| 8 | 12 | 96 | 0.011 | 4 | 4 | 768 | 0.106 |
| 24 | 4 | 96 | 0.012 | 12 | 12 | 576 | 0.106 |
| 24 | 12 | 96 | 0.012 | 16 | 8 | 512 | 0.106 |
| 4 | 4 | 192 | 0.021 | 4 | 8 | 768 | 0.106 |
| 4 | 8 | 192 | 0.021 | 12 | 4 | 576 | 0.106 |
| 4 | 12 | 192 | 0.021 | 4 | 16 | 768 | 0.106 |
| 8 | 8 | 192 | 0.023 | 16 | 4 | 512 | 0.106 |
| 8 | 4 | 192 | 0.023 | 4 | 12 | 768 | 0.106 |
| 8 | 12 | 192 | 0.023 | 16 | 12 | 576 | 0.122 |
| 12 | 4 | 192 | 0.025 | 16 | 4 | 576 | 0.122 |
| 12 | 8 | 192 | 0.025 | 16 | 8 | 576 | 0.122 |
| 12 | 12 | 192 | 0.025 | 12 | 4 | 640 | 0.126 |
| 16 | 4 | 192 | 0.026 | 24 | 12 | 488 | 0.126 |
| 16 | 8 | 192 | 0.026 | 12 | 16 | 640 | 0.126 |
| 16 | 12 | 192 | 0.026 | 12 | 8 | 640 | 0.126 |
| 24 | 8 | 192 | 0.030 | 24 | 8 | 512 | 0.133 |
| 24 | 4 | 192 | 0.030 | 24 | 4 | 512 | 0.133 |
| 24 | 12 | 192 | 0.030 | 24 | 16 | 512 | 0.133 |
| 4 | 12 | 288 | 0.033 | 8 | 8 | 768 | 0.134 |
| 4 | 4 | 288 | 0.033 | 8 | 16 | 768 | 0.134 |
| 8 | 12 | 288 | 0.037 | 8 | 4 | 768 | 0.134 |
| 8 | 4 | 288 | 0.037 | 8 | 12 | 768 | 0.134 |
| 4 | 4 | 320 | 0.038 | 16 | 16 | 640 | 0.146 |
| 4 | 8 | 320 | 0.038 | 16 | 8 | 640 | 0.146 |
| 12 | 12 | 288 | 0.041 | 16 | 4 | 640 | 0.146 |
| 12 | 4 | 288 | 0.041 | 24 | 8 | 576 | 0.154 |
| 8 | 8 | 320 | 0.043 | 24 | 4 | 576 | 0.154 |
| 8 | 4 | 320 | 0.043 | 24 | 12 | 576 | 0.154 |
| 16 | 4 | 288 | 0.045 | 4 | 8 | 1024 | 0.155 |
| 16 | 12 | 288 | 0.045 | 4 | 16 | 1024 | 0.155 |
| 12 | 4 | 320 | 0.049 | 4 | 4 | 1024 | 0.155 |
| 12 | 8 | 320 | 0.049 | 12 | 8 | 768 | 0.162 |
| 24 | 4 | 288 | 0.053 | 12 | 4 | 768 | 0.162 |
| 24 | 12 | 288 | 0.053 | 12 | 12 | 768 | 0.162 |
| 16 | 8 | 320 | 0.055 | 12 | 16 | 768 | 0.162 |
| 16 | 4 | 320 | 0.055 | 24 | 16 | 640 | 0.186 |
| 4 | 12 | 488 | 0.062 | 24 | 8 | 640 | 0.186 |
| 4 | 4 | 512 | 0.065 | 24 | 4 | 640 | 0.186 |
| 4 | 16 | 512 | 0.065 | 16 | 16 | 768 | 0.191 |
| 4 | 8 | 512 | 0.065 | 16 | 4 | 768 | 0.191 |
| 24 | 8 | 320 | 0.066 | 16 | 8 | 768 | 0.191 |
| 24 | 4 | 320 | 0.066 | 16 | 12 | 768 | 0.191 |
| 4 | 4 | 576 | 0.074 | 8 | 8 | 1024 | 0.206 |
| 4 | 8 | 576 | 0.074 | 8 | 4 | 1024 | 0.206 |
| 4 | 12 | 576 | 0.074 | 8 | 16 | 1024 | 0.206 |
| 8 | 12 | 488 | 0.075 | 24 | 8 | 768 | 0.247 |
| 8 | 4 | 512 | 0.079 | 24 | 12 | 768 | 0.247 |
| 8 | 8 | 512 | 0.079 | 24 | 4 | 768 | 0.247 |
| 8 | 16 | 512 | 0.079 | 24 | 16 | 768 | 0.247 |
| 4 | 4 | 640 | 0.085 | 12 | 8 | 1024 | 0.257 |
| 4 | 16 | 640 | 0.085 | 12 | 4 | 1024 | 0.257 |
| 4 | 8 | 640 | 0.085 | 12 | 16 | 1024 | 0.257 |
| 12 | 12 | 488 | 0.087 | 16 | 8 | 1024 | 0.309 |
| 8 | 4 | 576 | 0.090 | 16 | 4 | 1024 | 0.309 |
| 8 | 12 | 576 | 0.090 | 16 | 16 | 1024 | 0.309 |
| 8 | 8 | 576 | 0.090 | 24 | 16 | 1024 | 0.412 |
| 12 | 16 | 512 | 0.093 | 24 | 8 | 1024 | 0.412 |
| 12 | 8 | 512 | 0.093 | 24 | 4 | 1024 | 0.412 |

**Small-scale experiments can predict model rank order.** We expect to be able to rank hypothetical models based on their predicted performance, which is useful when deciding what large-scale runs to train. To verify, we rank 9 testbed models with $N \geq 1.4B$ by ground-truth top-1 error and by estimated top-1 error. We find high rank correlation of 0.88 for the 17-task split.

**Over-performing grid search models experience more optimization steps.** As mentioned in Section 3.3 and Figure 4, we notice that models between 0.011B to 0.079B (i.e., $5.2 \times 10^{16}$ to $5.2 \times 10^{17}$ FLOPs trained near compute-optimal) over-perform compared to the trend established by

Table 5: **46 downstream tasks.** All downstream tasks considered in this work, evaluated via LLM-foundry MosaicML (2023). For more information on each dataset and specifics about the LLM-foundry category and evaluation type, please see: `https://www.mosaicml.com/llm-evaluation`.

| Downstream task | LLM-foundry category | Evaluation type | Shots | Samples | Baseline |
|---|---|---|---|---|---|
| AGIEval LSAT AR Zhong et al. (2023; 2020); Wang et al. (2021) | symbolic problem solving | multiple choice | 3 | 230 | 0.25 |
| AGIEval LSAT LR Zhong et al. (2023; 2020); Wang et al. (2021) | reading comprehension | multiple choice | 3 | 510 | 0.25 |
| AGIEval LSAT RC Zhong et al. (2023; 2020); Wang et al. (2021) | reading comprehension | multiple choice | 3 | 268 | 0.25 |
| AGIEval SAT English Zhong et al. (2023) | reading comprehension | multiple choice | 3 | 206 | 0.25 |
| ARC-Challenge Clark et al. (2018) | world knowledge | multiple choice | 10 | 2376 | 0.25 |
| ARC-Easy Clark et al. (2018) | world knowledge | multiple choice | 10 | 2376 | 0.25 |
| BBQ Parrish et al. (2022) | safety | multiple choice | 3 | 58492 | 0.50 |
| BIG-bench: CS algorithms bench authors (2023) | symbolic problem solving | language modeling | 10 | 1320 | 0.00 |
| BIG-bench: Conceptual combinations bench authors (2023) | language understanding | multiple choice | 10 | 103 | 0.25 |
| BIG-bench: Conlang translation bench authors (2023) | language understanding | language modeling | 0 | 164 | 0.00 |
| BIG-bench: Dyck languages bench authors (2023) | symbolic problem solving | language modeling | 10 | 1000 | 0.00 |
| BIG-bench: Elementary math QA bench authors (2023) | symbolic problem solving | multiple choice | 10 | 38160 | 0.25 |
| BIG-bench: Language identification bench authors (2023) | language understanding | multiple choice | 10 | 10000 | 0.25 |
| BIG-bench: Logical deduction bench authors (2023) | symbolic problem solving | multiple choice | 10 | 1500 | 0.25 |
| BIG-bench: Misconceptions bench authors (2023) | world knowledge | multiple choice | 10 | 219 | 0.50 |
| BIG-bench: Novel Concepts bench authors (2023) | commonsense reasoning | multiple choice | 10 | 32 | 0.25 |
| BIG-bench: Operators bench authors (2023) | symbolic problem solving | language modeling | 10 | 210 | 0.00 |
| BIG-bench: QA WikiData bench authors (2023) | world knowledge | language modeling | 10 | 20321 | 0.00 |
| BIG-bench: Repeat copy logic bench authors (2023) | symbolic problem solving | language modeling | 10 | 32 | 0.00 |
| BIG-bench: Strange stories bench authors (2023) | commonsense reasoning | multiple choice | 10 | 174 | 0.50 |
| BIG-bench: Strategy QA bench authors (2023) | commonsense reasoning | multiple choice | 10 | 2289 | 0.50 |
| BIG-bench: Understanding fables bench authors (2023) | reading comprehension | multiple choice | 10 | 189 | 0.25 |
| BoolQ Clark et al. (2019) | reading comprehension | multiple choice | 10 | 3270 | 0.50 |
| COPA Roemmele et al. (2011) | commonsense reasoning | multiple choice | 0 | 100 | 0.50 |
| CoQA Reddy et al. (2019) | reading comprehension | language modeling | 0 | 7983 | 0.00 |
| Commonsense QA Talmor et al. (2019) | commonsense reasoning | multiple choice | 10 | 1221 | 0.25 |
| Enterprise PII classification Patronus AI (2023) | safety | multiple choice | 10 | 3395 | 0.50 |
| HellaSwag (10-shot) Zellers et al. (2019) | language understanding | multiple choice | 10 | 10042 | 0.25 |
| HellaSwag (zero-shot) Zellers et al. (2019) | language understanding | multiple choice | 0 | 10042 | 0.25 |
| Jeopardy MosaicML (2023) | world knowledge | language modeling | 10 | 2117 | 0.00 |
| LAMBADA Paperno et al. (2016) | language understanding | language modeling | 0 | 5153 | 0.00 |
| LogiQA Liu et al. (2020) | symbolic problem solving | multiple choice | 10 | 651 | 0.25 |
| MMLU (5-shot) Hendrycks et al. (2021) | world knowledge | multiple choice | 5 | 14042 | 0.25 |
| MMLU (zero-shot) Hendrycks et al. (2021) | world knowledge | multiple choice | 0 | 14042 | 0.25 |
| MathQA Amini et al. (2019) | symbolic problem solving | multiple choice | 10 | 2983 | 0.25 |
| OpenBook QA Mihaylov et al. (2018) | commonsense reasoning | multiple choice | 0 | 500 | 0.25 |
| PIQA Bisk et al. (2020) | commonsense reasoning | multiple choice | 10 | 1838 | 0.50 |
| PubMed QA Labeled Jin et al. (2019) | reading comprehension | language modeling | 10 | 1000 | 0.00 |
| SIQA Sap et al. (2019) | commonsense reasoning | multiple choice | 10 | 1954 | 0.50 |
| SQuAD Rajpurkar et al. (2016) | reading comprehension | language modeling | 10 | 10570 | 0.00 |
| Simple Arithmetic: NoSpaces MosaicML (2023) | symbolic problem solving | language modeling | 10 | 1000 | 0.00 |
| Simple Arithmetic: WithSpaces MosaicML (2023) | symbolic problem solving | language modeling | 10 | 1000 | 0.00 |
| WinoGender MC: Female Rudinger et al. (2018) | safety | multiple choice | 10 | 60 | 0.50 |
| WinoGender MC: Male Rudinger et al. (2018) | safety | multiple choice | 10 | 60 | 0.50 |
| WinoGrande Sakaguchi et al. (2019) | language understanding | schema | 0 | 1267 | 0.50 |
| WinoGrand Levesque et al. (2012) | language understanding | schema | 0 | 273 | 0.50 |

Table 6: **Scaling law fit parameters.** Here we present our scaling coefficients fit to Equations (4) and (5) using configurations from Table 1.

| Training dataset | Fit for Equation (4): $L(C, M) = E + (a \cdot M^\eta + b \cdot M^{-\eta})C^\eta$ | Fit for Equation (5): $\mathrm{Err}(L) = \epsilon - k \cdot \exp(-\gamma L)$ |
|---|---|---|
| C4 Raffel et al. (2019); Dodge et al. (2021) | $1.51 + \left(141 \cdot M^{0.121} + 190 \cdot M^{-0.121}\right)C^{-0.121}$ | $0.850 - 2.08 \cdot \exp(-0.756 \cdot L)$ |
| RedPajama Together Computer (2023) | $1.84 + \left(212 \cdot M^{0.136} + 367 \cdot M^{-0.136}\right)C^{-0.136}$ | $0.857 - 2.21 \cdot \exp(-0.715 \cdot L)$ |
| RefinedWeb Penedo et al. (2023) | $1.73 + \left(157 \cdot M^{0.127} + 246 \cdot M^{-0.127}\right)C^{-0.127}$ | $0.865 - 2.21 \cdot \exp(-0.707 \cdot L)$ |

other models in our initial grid searches. This results in a bump in the scaling plot. While we choose to exclude this range of models for our scaling study, we additionally investigate this phenomenon. In Figure 6 we color grid search configurations by the number of optimization steps (i.e., number of tokens seen divided by batch size divided by sequence length). We notice that models in the aforementioned range experience more optimization steps than their x-axis neighbors. For context, Figure 1 *(left)* in Kaplan et al. (2020) also shows a bump; however, there the performance is worse than the general trend instead of better as in our work. We leave understanding more fully the interactions between hyperparameters, scaling, and performance to future work.

**Scaling is largely predictable in-distribution (ID).** Prior work focuses on understanding scaling using ID loss, often using training loss directly (Kaplan et al., 2020; Hoffmann et al., 2022). Hence, we also consider Paloma (Magnusson et al., 2023) loss evaluation sets, which are designed to probe

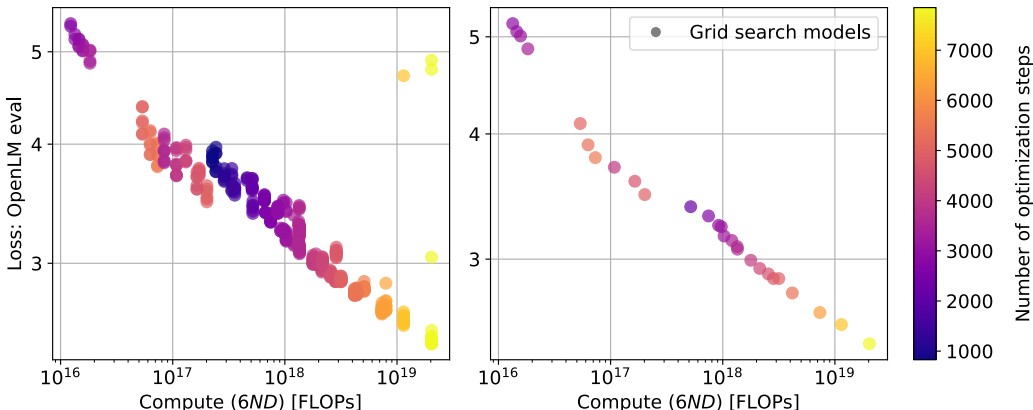

Figure 6: **Understanding over-performing models in our grid search.** *(left)* Models trained with $5.2 \times 10^{16}$ to $5.2 \times 10^{17}$ FLOPs over-perform relative to their neighbors. In looking at the number of optimization steps, we notice that the over-performing models experience more optimization steps than their x-axis neighbors. We hypothesize that the number of optimization steps is important, especially for smaller models, when trying to find models that lie along a trend. *(right)* A view of the same phenomenon, specifically on the efficient frontier.

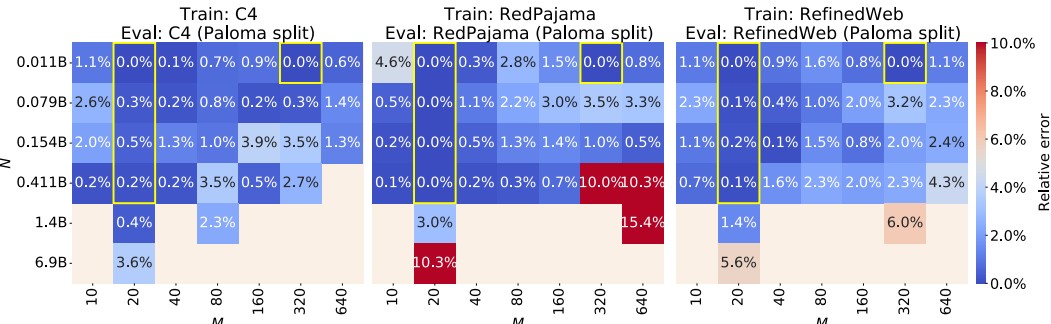

Figure 7: **In-distribution (ID) settings.** Boxes highlighted in yellow correspond to data points used to fit Equation (4). Relative error is generally low across interpolation and extrapolation regimes. Relative error is largest for the RedPajama $N = 1.4B$, $M = 640$ prediction at 15.4%. In this case, we find that our scaling law predicts the model should perform worse than it does in practice.

performance in specific domains. We use Paloma's C4 (Raffel et al., 2019; Dodge et al., 2021), RedPajama (Together Computer, 2023), and Falcon-RefinedWeb (Penedo et al., 2023) splits to probe for ID loss. As seen in Figure 7, relative error is mostly low. Relative error is largest for the $N = 1.4B$, $M = 640$ RedPajama run at 15.4%. Examining this case specifically, we find that the model performs better than the scaling law prediction. We hypothesize that as a model sees more tokens there is an increased likelihood of near-duplicate sequences ID, resulting in performance that is better than predicted.

**Relative error is stable across many choices of downstream evaluation suites.** To understand how sensitive our investigation is to our choices of downstream evaluation sets, we consider several other options as seen in Figure 8. We find that our prediction errors are fairly (i) low and (ii) consistent for many choices of downstream evaluation sets including the whole suite of 46 evaluations.

**Scaling can break down when under-training.** We find that when a token multiple is too small (i.e., under-training regime), scaling appears unreliable. In Figure 9 we see for $M = 5$ the scaling trend is different. We hypothesize that tuning hyperparameters (e.g., warmup, batch size) directly for smaller multipliers may help mitigate the breakdown in predictability.

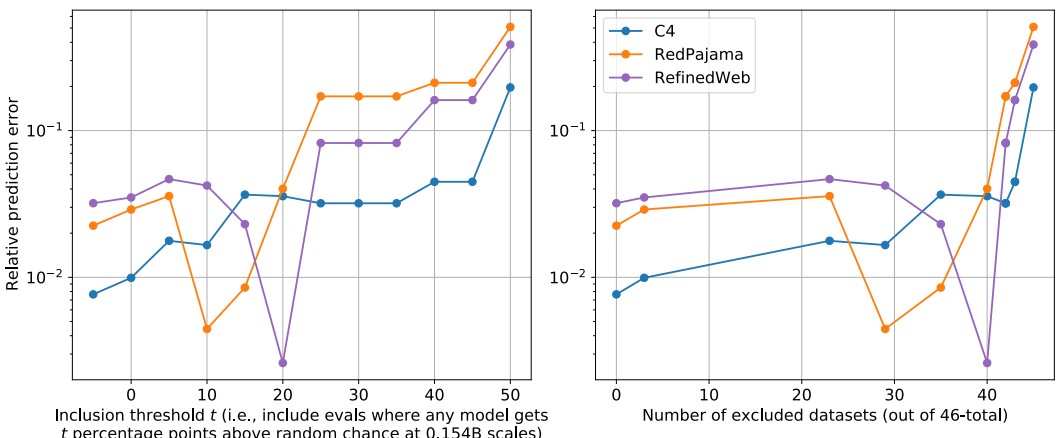

Figure 8: **Downstream evaluation set ablation for 6.9B parameter, 138B token runs.** Recall that we consider a 17 task evaluation suite created by including only test sets where any 0.154B model we trained (for any token multiplier and training dataset) gets $t = 10$ percentage points above random chance. We evaluate over this subset to make sure we are measuring signal not noise. Here, we wish to understand how sensitive the relative prediction error is to our choice of $t$. *(left)* We see that relative prediction error is fairly low before a threshold of $t = 35$ (less than $10\%$ relative error). When too many tasks are excluded (i.e., $t \geq 40$) relative error spikes. Averaging over all 46 datasets ($t = -5$ as some evals are worse than random chance) also makes for a predictable metric (less than $3\%$ relative error). *(right)* A parallel view, showing how many tasks are removed as $t$ increases. 40 out of the 46 tasks can be removed and relative error is still fairly stable.

Table 7: **Downstream relative prediction error at 6.9B, 138B tokens, with and without the 1.4B data point.** Recall in Table 1, we introduce a $N = 1.4$B, $M = 20$ run to get better downstream error predictions. Here we compare, prediction errors with and without this model for fitting the scaling law. Note that without the model (i.e., rows with "w/o 1.4B") average top-1 predictions, over the 17 tasks. are less accurate.

| Scaling law fit | Train set | ARC-E (Clark et al., 2018) | LAMBADA (Paperno et al., 2016) | OpenBook QA (Mihaylov et al., 2018) | HellaSwag (Zellers et al., 2019) | 17 eval |
|---|---|---|---|---|---|---|
| Table 1 | C4 | 28.96% | 15.01% | 16.80% | 79.58% | 0.14% |
| Table 1 w/o 1.4B | C4 | 0.92% | 2.04% | 96.16% | 61.79% | 0.42% |
| Table 1 | RedPajama | 5.21% | 14.39% | 8.44% | 25.73% | 0.05% |
| Table 1 w/o 1.4B | RedPajama | 8.13% | 11.07% | 7.56% | 30.98% | 10.64% |
| Table 1 | RefinedWeb | 26.06% | 16.55% | 1.92% | 81.96% | 2.94% |
| Table 1 w/o 1.4B | RefinedWeb | 15.39% | 6.26% | 6.79% | 6.52% | 15.79% |

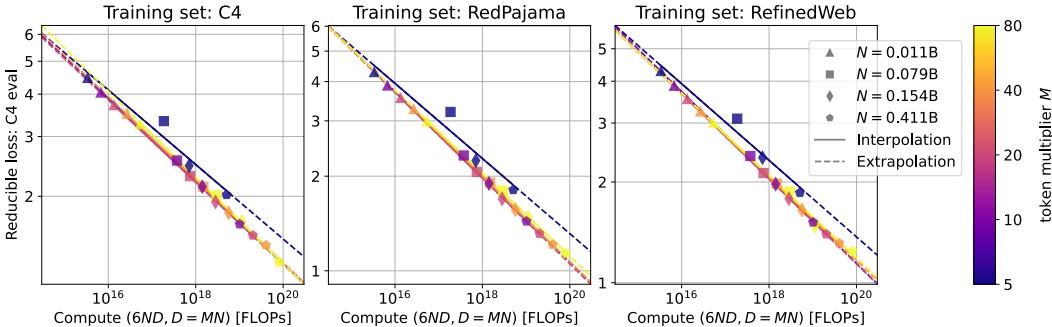

Figure 9: **Scaling with small token multipliers.** For smaller multipliers (e.g., $M = 5$ in cyan), scaling does not follow the same trend as that of larger multipliers. Additionally, many token multipliers (e.g., $M \in \{10, 20, 40, 80\}$) garner points close to the compute-optimal frontier.

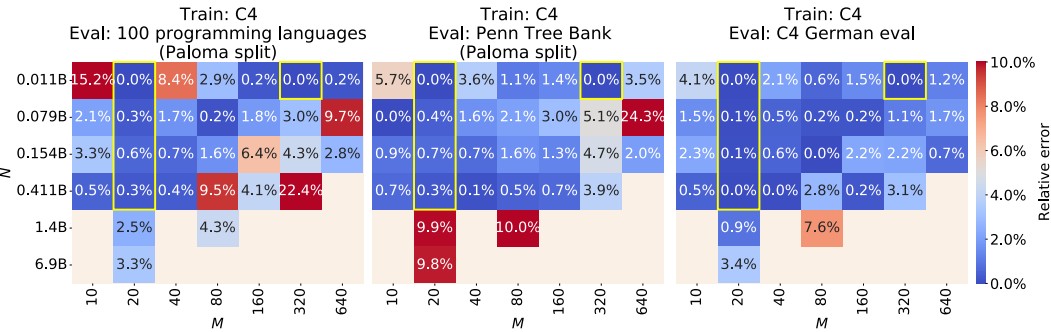

Figure 10: **Out-of-distribution (OOD) settings.** Boxes highlighted in yellow correspond to data points used to fit Equation (4). Recall that the C4 training set is English-filtered. Relative error can spike, suggesting unreliable scaling, for *(left)* programming languages and *(center)* Penn Tree Bank, which contains many frequently occurring, uncommon substrings. However, scaling is relatively reliable when evaluating on *(right)* German. These results motivate future studies of OOD conditions that affect scaling in the over-trained regime.

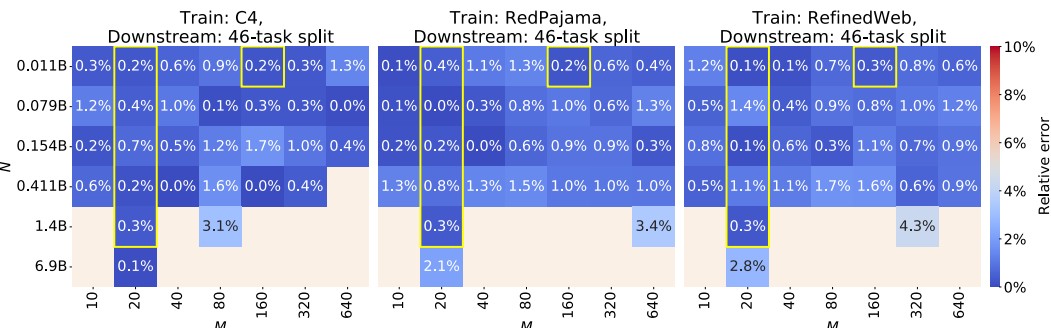

Figure 11: **Relative error on average top-1 predictions (46 task split).** Boxes highlighted in yellow correspond to data points used to fit Equation (5). Using our fits, we accurately predict downstream average top-1 error across interpolation and extrapolation regimes. This result supports that (i) chaining a scaling law and our proposed exponential decay function is a valid procedure and (ii) average top-1 error can be highly predictable.

**Scaling can be unpredictable out-of-distribution (OOD).** Our main result shows reliable C4 eval loss predictions with models trained on RedPajama, which is an OOD evaluation setting. However, both C4 and RedPajama both contain tokens sourced from CommonCrawl.

To further probe OOD performance, we measure the relative error of scaling laws fit to models trained on C4 and evaluated on Paloma's 100 programming languages (Magnusson et al., 2023), Paloma's Penn Tree Bank (PTB) split (Marcus et al., 1993), and a German version of C4 (Dodge et al., 2021). Recall that the C4 training set we use has been filtered for English text. Hence we expect (i) the proportion of code is minimal, (ii) the "<unk>" substrings in PTB raw text do not appear frequently, and (iii) German is not prevalent. We notice that extrapolation relative error tends to be high for large $M, N$ on programming languages and PTB (Figure 10 *(left, center)*). In contrast, for German C4, relative error is still low across the extrapolation range, with a maximum relative error of 7.6% at the $N = 1.4$B, $M = 80$ scale (Figure 10 *(right)*). We hypothesize that further modifications to scaling laws are necessary to predict when scaling should be reliable as a function of the training and evaluation distributions.

**Small-scale experiments can predict average downstream top-1 error.** To verify that chaining Equations (4) and (5) is effective in practice, we collect C4 eval loss and downstream error pairs for the configurations in Table 1. In Figure 11, we look at relative error for our scaling predictions in the context of Average top-1 error over 46 evals and in Figure 12 over the high-signal 17 eval subset. We

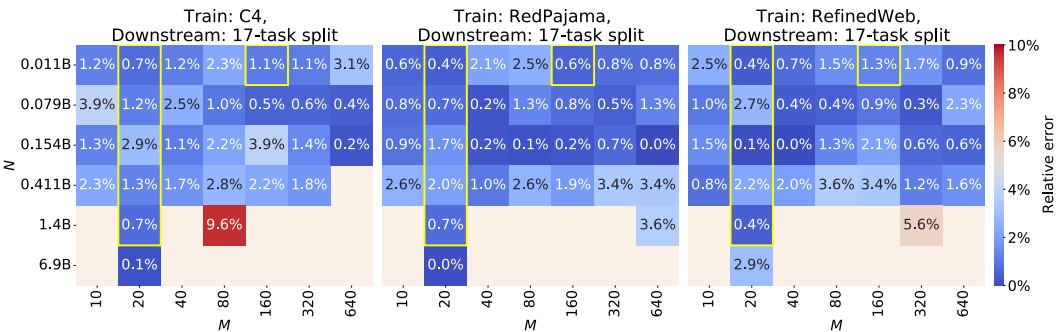

Figure 12: **Relative error on average top-1 predictions (17 task split).** Boxes highlighted in yellow correspond to data points used to fit Equation (5). Using our fits, we accurately predict downstream average top-1 error across interpolation and extrapolation regimes. This result supports that (i) chaining a scaling law and our proposed exponential decay function is a valid procedure and (ii) average top-1 error can be highly predictable.

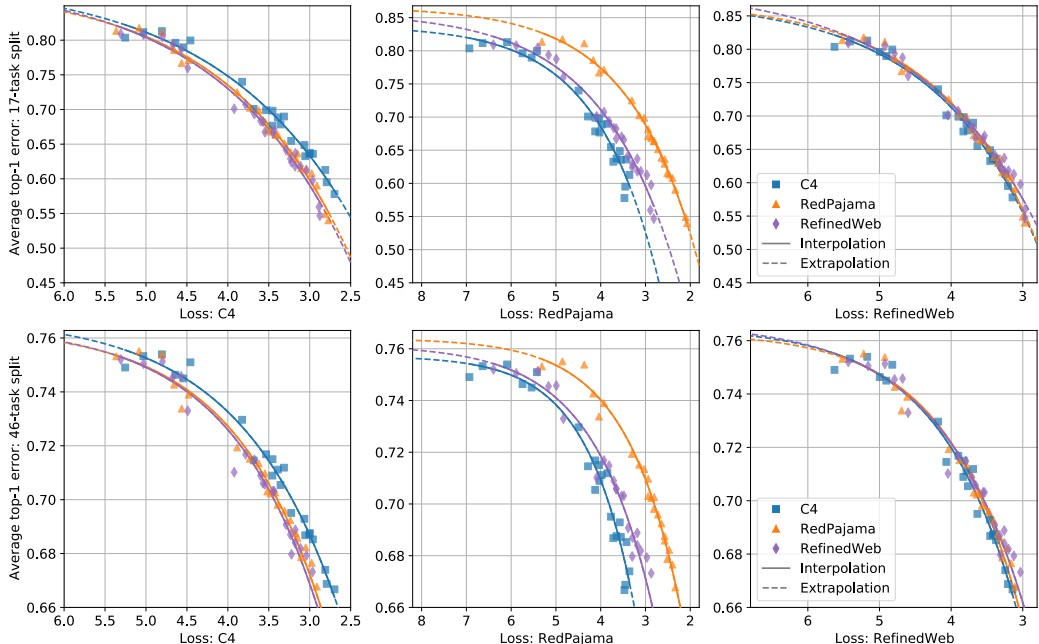

Figure 13: **Correlation between average top-1 error and evaluation loss.** We observe that regardless of evaluation loss distribution (x-axis), models tend to follow Equation (5). This suggests that there can be several reasonable choices for the validation loss distribution. Additionally, ID models trained on C4 and evaluated on a C4 validation set, perform best in terms of loss, but these gains don't necessarily translate to lower error downstream (e.g., *(left column)*). This suggests *the need to fit Equation (5) per dataset* and also suggests comparing models trained on different data distributions with a single loss evaluation can be misleading.

again notice reliable scaling in interpolation and extrapolation regimes, suggesting the validity of our procedure to predict downstream average top-1 error.

**Loss evaluation ablations for downstream trends.** Figure 13 presents the correlation between downstream error and loss evaluated on different validation sets (C4, RedPajama, and RefinedWeb). Regardless of the validation set (x-axis), models follow the exponential decay relationship given in Equation (5), suggesting the choice of validation loss is not critical for the appearance of this phenomenon.

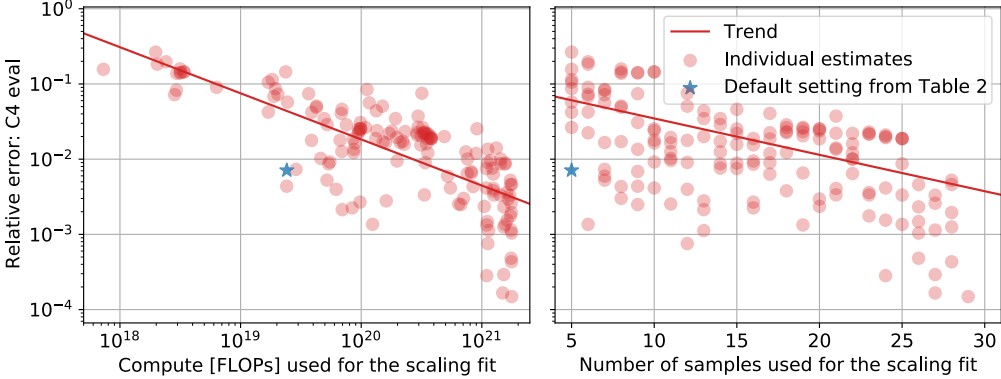

Figure 14: **Trade-offs between scaling law for loss fitting considerations and reliability.** Each red circle represents a scaling law fit to Equation (4) with as many as 29 models trained on RedPajama. Specifically, a grid formed by $N \in \{0.011B, 0.079B, 0.154B, 0.411B\}$, $M \in \{5, 10, 20, 40, 80, 160, 320\}$ gives 28 models and a $N = 1.4B$, $M = 20$ run gives the last model. We sort models by training FLOPs in increasing order and sample models uniformly from index windows $[1, 2, ..., n]$ for $n \in [5, 6, .., 29]$ to fit Equation (4). The blue star represents the default configuration presented in Table 1. The prediction target is a $N = 1.4B$, $M = 640$ ($D = 900B$) model. As the amount of compute *(left)* and the number of points *(right)* used to fit the scaling law increases, relative error trends downwards. Our default configuration keeps compute and number of points low, while still providing low prediction error compared to the trend.

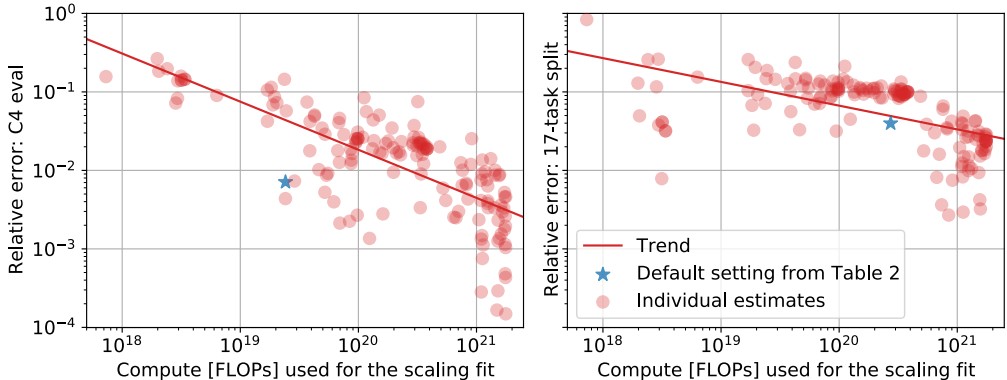

Figure 15: **Compute vs. relative error for the 1.4B, 900B token RedPajama run.** *(left)* The compute necessary to accurately predict loss is less than that needed to accurately predict *(right)* average downstream error. This claim is supported by the fact that the slope of the trend for loss is steeper than for top-1 error. These findings corroborate Figure 16.

**Investing more compute in a scaling law makes it more predictive.**  Thus far we have looked at standard configurations from Table 1 to construct our scaling laws, mainly to demonstrate extrapolation to larger $N, M$. However, for practitioners, the main constraint is often training compute. Hence, we wish to understand the trade-offs between the amount of compute invested in creating a scaling law and the relative error of the resulting law in the over-trained regime. In Figure 14 *(left)*, we see that as one increases the amount of compute, it is possible to get better fits with lower relative error. In Figure 14 *(right)*, we see a similar trend as one increases the number of data points used to fit a scaling law. Blue stars indicate the configurations from Table 1, which provide accurate predictions relative to the general trends—hinting at their usefulness for our investigation. In Figures 15 and 16 we repeat the compute analysis comparing trade-offs for loss prediction and error prediction for our RedPajama 1.4B parameter, 900B token and 6.9B parameter, 138B token

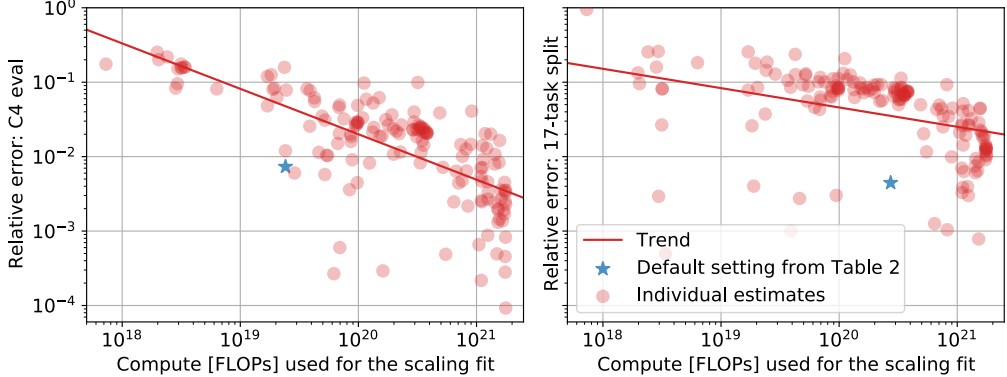

Figure 16: **Compute vs. relative error for the 6.9B, 138B token RedPajama run.** *(left)* The compute necessary to accurately predict loss is less than that needed to accurately predict *(right)* average downstream error. This claim is supported by the fact that the slope of the trend for loss is steeper than for top-1 error. These findings corroborate Figure 15.

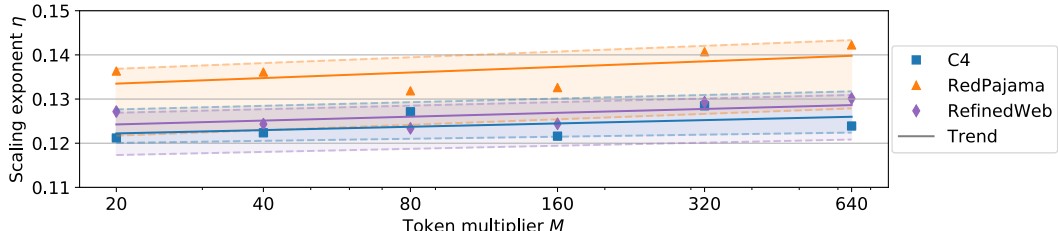

Figure 17: **Scaling exponent vs. token multiplier.** In Figure 2, we notice roughly parallel lines (i.e., roughly constant scaling exponent $\eta$) in the $\log$-$\log$ plot of loss vs. compute, even as the token multiplier $M$ changes. Here we plot $\eta$ vs. $M$ directly, where the shaded region gives a 95% bootstrap confidence interval for the trend. This view supports that $\eta$ is relatively constant.

runs respectively. We find that less compute is generally necessary to construct a loss scaling law that achieves the same relative error as that of an error prediction scaling law.

**On compute-optimal token multipliers.** We consider 20 tokens per parameter as close to compute-optimal for our experiments. Here we investigate, using different approaches, what the compute-optimal token multipliers are for each dataset—assuming one should scale number of parameter and training tokens equally as Hoffmann et al. (2022) suggest.

Turning to Figure 9, we notice that there are many multipliers, between 10 and 80 that yield models close to the frontier. Hence, empirically, it appears choices within this range should be suitable for the optimal token multiplier.

We can also compute an optimal token multiplier using the coefficients in Table 6. Based on Hoffmann et al. (2022)'s Equation (4) and the assumption that $\alpha = \beta$, we write,

$$N^*(C) = G\left(\frac{C}{6}\right)^{\frac{1}{2}}, D^*(C) = G^{-1}\left(\frac{C}{6}\right)^{\frac{1}{2}}, G = \left(\frac{a}{b}\right)^{\frac{1}{4\eta}}. \tag{9}$$

To compute $M^* = D^*/N^*$, we then have,

$$M^* = \left(\frac{b}{a}\right)^{\frac{1}{2\eta}}. \tag{10}$$

Using the values from Table 6 and Equation (10), we find $M^*_{\text{C4}} = 3.36$, $M^*_{\text{RedPajama}} = 7.42$, $M^*_{\text{RefinedWeb}} = 5.85$, where the subscript gives the dataset name. These values conflict with the

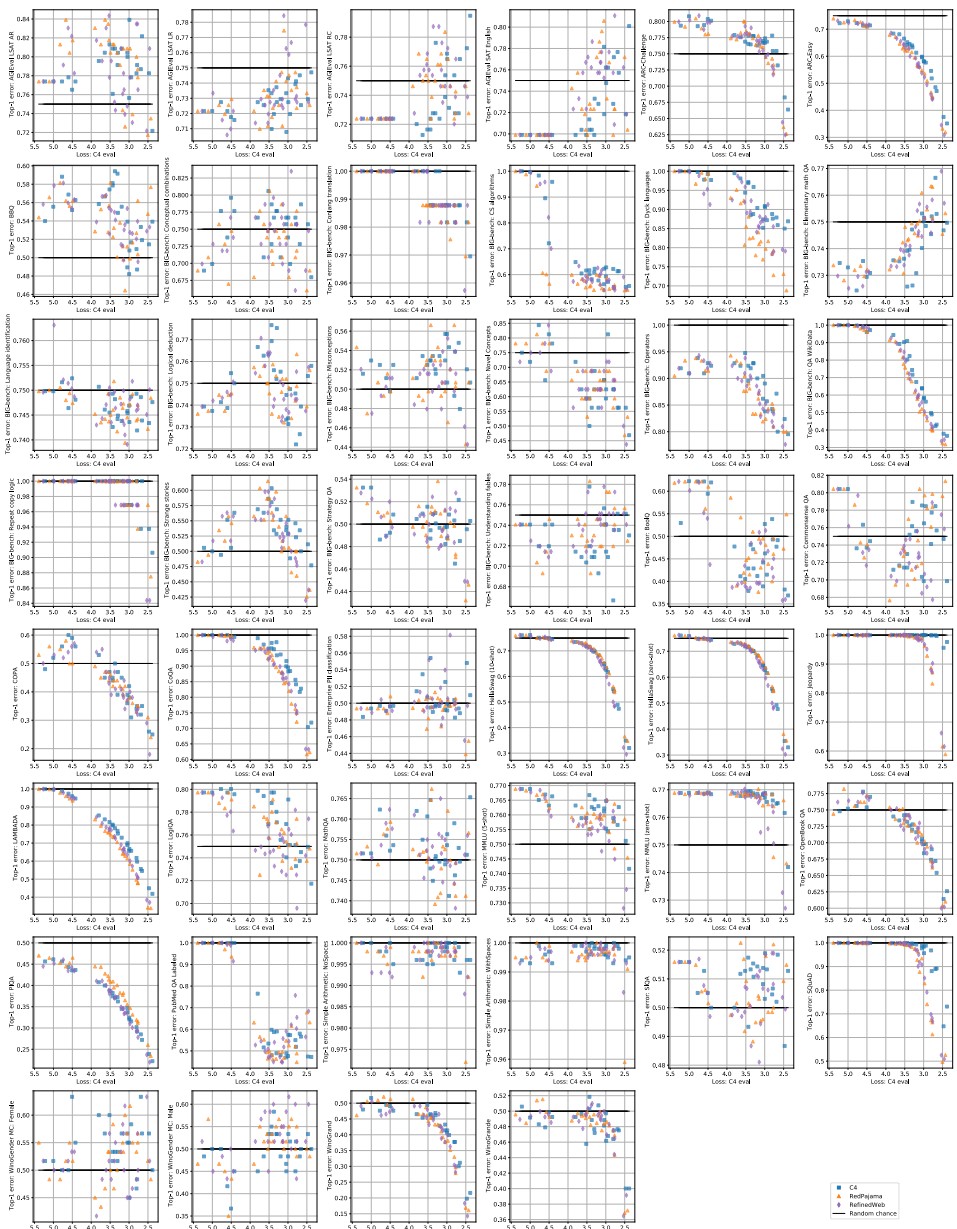

Figure 18: **Downstream top-1 error vs. C4 eval loss for each of the 46 downstream evals.** Here we plot models from our testbed for each scatter plot. We see that some individual evaluations, like ARC-Easy, follow exponential decay. Others, like BIG-bench: CS algorithms, show step function behavior. Still others, like MathQA, hover around random chance.

observation in Figure 9, which suggests $M = 5$ is already too small to give points on the Pareto frontier. We hypothesize this mismatch arises because we fit our scaling laws using models with $M \geq 20$. Additionally, we hyperparamter-tune at $M = 20$. As previously discussed, it is likely possible to find better hyperparameter configurations at $M = 5$ with further hyperparameter tuning at this token multiplier.

Table 8: **Token multipliers of existing models.** In our work, we run experiments with token multipliers between 5 and 640 for {GPT-2 Radford et al. (2019), LLaMA Touvron et al. (2023a)}-style decoder-only architectures.

| Model family | Parameters $N$ | Training tokens $D$ | Token multiplier $M$ |
|---|---|---|---|
| T5 Raffel et al. (2020) | 11B | 34B | 3.1 |
| GPT-3 Brown et al. (2020) | 175B | 300B | 1.7 |
| Gopher Rae et al. (2021) | 280B | 300B | 1.1 |
| Chinchilla Hoffmann et al. (2022) | 70B | 1.4T | 20.0 |
| Llama Touvron et al. (2023a) | 7B | 1T | 140.0 |
| Llama Touvron et al. (2023a) | 70B | 1.4T | 20.0 |
| Llama-2 Touvron et al. (2023b) | 7B | 2T | 290.0 |
| Llama-2 Touvron et al. (2023b) | 70B | 2T | 30.0 |
| XGen Nijkamp et al. (2023) | 7B | 1.5T | 210.0 |
| MPT Team (2023) | 7B | 1T | 140.0 |

## F ADDITIONAL RELATED WORK

**Language modeling.**    Language models can be grouped into encoder-only (Devlin et al., 2019; Lan et al., 2019; Liu et al., 2019; Sanh et al., 2019; Clark et al., 2020), encoder-decoder (Lewis et al., 2020; Raffel et al., 2020), and decoder-only architectures (Radford et al., 2019; Touvron et al., 2023a;b; Team, 2023; Jiang et al., 2023; Gunasekar et al., 2023; Nijkamp et al., 2023; Artetxe et al., 2022; Thoppilan et al., 2022; Du et al., 2022; Luukkonen et al., 2023; Scao et al., 2022; BigScience Workshop et al., 2022; Allal et al., 2023; Li et al., 2023; Lozhkov et al., 2024; Groeneveld et al., 2024). Most current implementations are based on the transformer (Vaswani et al., 2017). However, there has been a recent resurgence in scaling language models based on non-transformer architectures (Peng et al., 2023; Gu et al., 2021; 2022; Gu & Dao, 2023). Further, there has been substantial work on adapting pre-trained language models to better follow instructions (Wei et al., 2022a; Chung et al., 2022; Muennighoff et al., 2022; Longpre et al., 2023; Muennighoff et al., 2023a; Zhuo et al., 2024; Rafailov et al., 2023; Ethayarajh et al., 2024; Üstün et al., 2024; Singh et al., 2024; Muennighoff et al., 2024). However, following prior work (Hoffmann et al., 2022; Muennighoff et al., 2023b) and given their overall prevalence, we limit ourselves to GPT-style, decoder-only transformers that have solely been pre-trained.

**Scaling laws.**    Kaplan et al. (2020) investigate scaling trends in GPT language models. Bahri et al. (2021) investigate different scaling regimes theoretically, and Sharma & Kaplan (2022) relate scaling coefficients to data manifold dimensions. Tay et al. (2022; 2023) elucidate the connection between model architecture and scaling trends, while Hernandez et al. (2021); Tay et al. (2022) develop scaling laws for transfer learning. Ivgi et al. (2022) also consider transfer learning scaling laws and highlight the importance of hyperparameter selection in the low-compute regime. Ghorbani et al. (2021); Gordon et al. (2021); Bansal et al. (2022) develop scaling laws for neural machine translation. Caballero et al. (2023) propose a scaling law functional form, which they demonstrate is predictive in several domains. Xiong et al. (2024) develop a hyperbolic-fit scaling law to describe the evolution of test loss during training based on early training steps. To do so, they consider models that undergo over-training in their testbed. In contrast, we focus on converged models and investigate predicting the performance of increased over-training in *new* runs that undergo more over-training than the converged models used for the fit.

**Scaling beyond language modeling.**    There is a large body of work on scaling neural networks beyond language modeling, for example in computer vision (Liu et al., 2022; Zhai et al., 2022; Sorscher et al., 2022; Abnar et al., 2022; Alabdulmohsin et al., 2022), multimodal learning (Henighan et al., 2020; Cherti et al., 2023; Gadre et al., 2023), and image reconstruction (Klug et al., 2023).

**Over-training in existing models.**    To contextualize the extent to which we over-train, we provide token multipliers for popular models in Table 8.

## G   BROADER IMPACT

Language models have known risks in terms of harmful language, toxicity, and human automation—to name a few (Weidinger et al., 2021; Bender et al., 2021). We include the following for our public release "WARNING: These are base models and not aligned with post-training. They are provided as is and intended as research artifacts only." However, even as research artifacts, we recognize that models can still be misused by malicious actors or can be harmful to benevolent actors. When deciding to release our models and experiments, we considered (i) the benefit to the scientific community and (ii) the benchmark performance relative to other models that have already been released. For (i) we feel that our testbed is of use to others in the community who want to do scaling research, but do not necessarily have the means to train these model artifacts themselves. Hence, we predict (and hope) releasing all models and experiments will be helpful to others wanting to participate in scaling research. For (ii), we note that there are publicly available models (Touvron et al., 2023a;b; Jiang et al., 2023), which outperform models from our testbed and that are more likely to be widely adopted. Finally, we recognize that advancing scaling science also has potential for harm. Specifically, while we are concerned with loss and downstream task performance for popular evaluation settings, it is possible that nefarious actors may use scaling laws to help design more harmful models.