# OpenReview forum: "Language models scale reliably with over-training and on downstream tasks"
_ICLR.cc/2025/Conference — ICLR 2025 Poster_

### Official Review · Reviewer_HT5J · 2024-10-29

**Soundness:** 4
**Presentation:** 4
**Contribution:** 4
**Rating:** 8
**Confidence:** 5

**Summary:**

This article investigates the behaviors of scaling laws in the regime of over-training and on downstream benchmarks. Through derivation and initial observations, the article provide empirical evidence that over-training largely follows a similar, tractable, and constant-slope scaling law. Through further investigations and experiments, the article derives the parameters in the scaling law for over-training. With that conclusions, the article is then able to derive a scaling law relating model size, training tokens, and average downstream task performance, for the first time proving that average performance across a subset of downstream tasks are tractable.

**Strengths:**

1. Originality: This work firstly investigates and tracks next-token prediction model performance under the over-training regime and on downstream benchmark evaluation tasks.
2. Strong experiment results: This work builds a test-bed with more than 100 models with different sizes and training tokens. Through predicting the scaling law, this work successfully predicts the performance of larger models trained with more tokens.
3. Theoretical relationship with prior work: This work provides a stimulating analysis (Section 2) between scaling laws under the over-training regime and existing Chinchilla/Kaplan scaling laws, providing strong theoretical evidence supporting the derived over-training scaling law.
4. Writing: This paper is well-written with a clear, logical presentation of motivation, derivation process, experiment result analysis, and conclusion.

**Weaknesses:**

1. This work suggests that "there remain gaps between current scaling studies and how language models are ultimately trained and evaluated" (Line 13-14), suggesting that over-trained regime is under-investigated. There is an interesting work [1] that investigates the pre-training process of language models. In [1], the authors predict the training outcomes using the gathered data from an initial training period, and they successfully predicts the training outcomes of relatively small models (<0.1B) trained on large number of tokens (400B tokens). In that case, $M>4000$ and falls into the over-training category. It would be interesting to investigate the difference between the proposed over-train scaling law and [1].

[1] Temporal Scaling Law for Large Language Models

**Questions:**

There is only one thing that WILL NOT affect my overall rating towards this article.

1. See Weakness [1], it is better to discuss the differences between the proposed over-train scaling law and [1].

I am willing to defend my overall rating in the rebuttal period.

[1] Temporal Scaling Law for Large Language Models

---

> ### Author Response · Authors · 2024-11-25
>
> **Relation to Xiong et al. [1].** Thank you for mentioning the reference! We were not aware of this paper at the time of our submission; however, have added the following sentence to our revised manuscript in our extended related work:
>
> >Xiong et al. (2024) develop a hyperbolic-fit scaling law to describe the evolution of test loss during training based on early training steps. To do so, they consider models that undergo over-training in their testbed. In contrast, we focus on converged models and investigate predicting the performance of increased over-training in *new* runs that undergo more over-training than the converged models used for the fit.
>
> We thank the reviewer for their generous review, acknowledging our originality, experimentation, contextualization relative to prior work, and presentation. Thanks for your attention to our work and for your willingness to defend your review during the review process!

---

> > ### Comment · Reviewer_HT5J · 2024-11-26
> >
> > Many thanks to the authors for their response. I will keep my overall rating even after reading other reviewer's comments. From my perspect of view, this is an inpiring paper and should be highlighted by the research community.

---

### Official Review · Reviewer_tXac · 2024-11-03

**Soundness:** 3
**Presentation:** 3
**Contribution:** 2
**Rating:** 6
**Confidence:** 3

**Summary:**

This paper explores the implications of over-training in LLMs, and tries to fit a scaling law for extrapolating beyond the compute-optimal (Chinchilla optimal) regime. The proposed scaling law is able to predict the final validation loss of a model using 300x less compute than what would have been needed to fully train the model.

**Strengths:**

- Extensive experiments across many training datasets (C4, RedPajama and RefinedWeb) and model sizes.
- Comprehensive evaluations using diverse validation datasets.

**Weaknesses:**

- As the field of LLMs is currently agressively expanding into multimodality (images, audio, video and data), the proposed method might soon not hold up anymore. There is no mention of early-fusion multimodal models or token-free language modeling.
- Lack of consideration for post-training, fine-tuning and inference-time compute scaling methods, which are now integral to all production-ready models.

**Questions:**

- How accurate would this method be for predicting the performance of much larger LLMs, such as 30B, 70B, 400B? Are there any special considerations in order to ensure the law stays accurate?
- Would this law also work on "efficient" LM architectures such as mixture-of-experts, approximate attention and attention-free models?
- How might overtraining affect post-training, fine-tuning or inference-time compute scaling methods? Would their benefits scale proportionally the same to the losses and evals?

---

> ### Author Response · Authors · 2024-11-25
>
> **Scaling laws for multimodal models, mixture-of-experts etc.** Thanks for bringing this up! We agree that understanding the scaling behavior of multimodal models is interesting; however, we consider it outside the scope of this paper, which focuses on language modeling. One benefit of studying scaling in the context of language modeling is that pre-training recipes have largely converged in this area of research (i.e., similar to GPT-2 and Llama training recipes). While there are exciting developments with sparse, mixture-of-experts (MoE), models, there are still fundamental design decisions that remain unclear (e.g., how best to route forward passes in MoE architectures). Given these uncertainties in emerging model families, we decided to focus computational resources on dense decoder-only transformers, which have had staying power in modern research and application.
>
> **Considering post-training.** Thank you for bringing up post-training! This is indeed an exciting area for future work with respect to scaling laws as we identify in our conclusion. The focus of our study is on pre-training, in line with prior work on model scaling (i.a, Chinchilla). The field of scaling laws is still maturing and we hope others might build on our testbed of >100 models to investigate scaling in post-training. Considering a small model that is over-trained to match the performance of a larger model; we hypothesize that the large model will be easier to post-train given the additional capacity from having more weights. However, because post-training is a relatively cheap operation compared to pre-training; we hypothesize also that differences between the aforementioned small model and large model may diminish as one scales up pre-training FLOPs keeping post-training FLOPs constant.
>
> **Considerations for predicting performance of larger language models.** There are a few considerations here, which we feel are important. Thanks for the opportunity to share! 1) Task difficulty. Once models become extremely capable, they tend to saturate performance metrics, specifically for easier tasks. When considering models up to 400B parameters, it probably makes sense to include more difficult tasks where performance is not saturated, even at this scale. 2) Knowledge of training data and multi-epoched effects. The scaling laws developed in this paper require knowledge of the training distribution. This is not known for highly capable closed source models or even for open-weight models such as those in the Llama3 family. Hence, it is hard to construct small scale proxies here. Also as one gets to high token training regimes, it is increasingly likely that one may have to train on repeated data (i.e., in a multi-epoch setting). It is likely that such settings affect scaling laws as supported by Muennighoff et al., 2023b.

---

### Official Review · Reviewer_fbs9 · 2024-11-04

**Soundness:** 3
**Presentation:** 2
**Contribution:** 3
**Rating:** 6
**Confidence:** 3

**Summary:**

The paper investigates scaling laws for language models  (LM) in the over-trained regime, where models are trained with greater token-to-parameter ratios to lower inference costs.
They furthermore derive power-law scaling laws for downstream task performance (error rates) from the conventional laws using LM loss.
Using a testbed of 104 models (0.011B to 6.9B parameters) with varying token counts across three distinct datasets, they fit power-law scaling laws to predict validation loss and error rates, validating the consistency and reliability of the proposed laws.
The paper is well-motivated and addresses an important problem, with technically significant and practically valuable findings.

**Strengths:**

* The problem of scaling laws under over-training is practical and interesting, addressing gaps in existing literature around compute-optimal training, allowing for the efficient resource utilization in large-scale language modeling.
* The proposed laws appear logical
* The evaluation appears comprehensive, with a diverse and appropriate testbed to test the hypotheses around scaling laws under over-training. As a result, they achieved reasonably good performance: average top-1 error on 17 tasks within a 0.05% relative error and 0.7% relative error on validation loss.
* The paper is clearly presented, with detailed steps for law derivation,  model configuration, and scaling law fitting.

**Weaknesses:**

- Performance of downstream task: Although average performance aligns with the laws, variability in individual predictions and occasional high errors may suggest that the scaling laws may not universally hold without further model or data refinements. Furthermore, the laws in equation 4 assume $\alpha =\beta$ which can limit the applicability.
- The paper does not address how the scaling laws would hold under different architectural configurations or training optimizations, which are relevant to future model development.
- The paper explored limitations such as sensitivity in out-of-distribution settings, posing challenges for future refinement of scaling laws in varied training regimes and tasks.
- The claim of "predictable downstream performance" seems somewhat misleading, as the performance for specific tasks can vary widely. A more precise phrasing could emphasize “predictable average performance” to better align with the actual findings.
- Presentation:
1. In Figure 1, the paper defines M = N/D (line 70) while D=MN (Fig 1 caption) and line 127
2. Figures need better visualization and explanation for understanding. For example, in Figure 1, colors for M=320 and M=640 are very similar.
3. Typos: Line 142: than in

**Questions:**

* Lines 142-144: can you elaborate on this sentence: "While loss should be higher than in the compute-optimal allocation for a given training budget, the resulting models have fewer parameters and thus incur less inference cost." and why it is "common practice to over-train smaller models" (line 44)?
* The exponential relationship between loss and error is motivated in Figure 3, but have you evaluated larger sets of tasks to validate the relationship?
* Does the paper assume that over-training effects remain uniform across model sizes and datasets?
* Could the authors elaborate on the characteristics of tasks where downstream predictions were less accurate? Are there specific aspects that tend to yield higher prediction errors?
* As the paper mentions exploring out-of-distribution scaling, can you provide findings on how well the scaling laws hold when tested on domains or tasks different from the training dataset?

---

> ### Author Response · Authors · 2024-11-25
>
> **Individual task prediction and assumptions.** Our work corroborates that of Schaeffer et al. (2023) that individual task prediction is indeed a hard problem. We identify this as a critical area for future study in our conclusion. Also, thanks for bringing up the $\alpha = \beta$ assumption. In Appendix A, we discuss extensions of the over-training scaling law in cases where  $\alpha \neq \beta$, finding that similarly predictable behavior is expected. However, given the large compute requirements needed to train models, we were unfortunately only able to restrict our empirical setup to the $\alpha = \beta$ regime.
>
> **Additional training recipes.** While we conducted experiments with various hypers and datasets, we acknowledge the limitation of not exploring additional optimizers or architectures. We feel our choice is justified, based on the standardization in modern decoder-only transformer training recipes (i.a., GPT-2, Llama). We recognize that focusing solely on the current training paradigm may be restrictive. However, the substantial compute requirements for a comprehensive scaling study pose practical challenges to broader exploration.
>
> **On predictable average downstream performance.** We are sorry for any confusion here! Our claim is indeed that average performance can be predictable. We try to be clear in our abstract (L025-026), intro (L091-092), and experiments (L413). However, if there are any specific instances where our claims are imprecise, we are happy to update the manuscript!
>
> **Typos.** We have updated the manuscript accordingly, thanks!
>
> **Figure readability.** Thanks for the suggestion. We updated Figure 1, 2, 9 and are happy to make further changes.
>
> **Elaboration on motivation for over-training (re: L142-144, 44).** Thanks for the chance to clarify! The fact that over-training reduces inference cost is the primary motivation for over-training in practice. Consider a Chinchilla optimal model $A$ that has $N$ parameters and is trained for $D$ tokens. This model takes approximately $6 \cdot N \cdot D$ FLOPs of compute to train. Now consider a second model $B$ that has only $N/c$ parameters, where $c>1$, but that is trained on $k \cdot c \cdot D$ tokens, where also $k>1$. This model takes $6 \cdot k \cdot N \cdot D > 6 \cdot N \cdot D$ FLOPs (i.e., it is more expensive to train). Consider that $B$ is over-trained s.t. it matches $A$ in validation loss (i.e., performance). Model $B$ is *cheaper* to serve at inference, as it has fewer parameters and requires less inference FLOPs.
>
> **Larger sets of tasks to validate the relationship between loss and downstream error.** Please see Figure 8 in the Appendix where we look at the relationship between loss and downstream performance for as many as 46 different tasks. The high level takeaway is that adding more evaluation tasks keeps the scaling laws relatively stable in their predictive power. However, considering too few tasks in the average can diminish predictive power.
>
> **Assumptions about the uniformity of over-training effects over datasets and model sizes.** Thanks for bringing this up! Our assumptions here are baked into Equation (4); namely, that the variables for the fit are sufficient to describe behavior. This equation is fit differently for different training distributions, so we do not assume that one scaling fit will be able to predict performance for models trained on a different training distributions.
>
> **Trends in individual task prediction error.** Looking at Table 2, it appears there are not clear trends in what individual tasks are predictable. Furthermore, it appears that predictability can differ wildly based on the *training distribution*. Particularly, we believe that this observation motivates future work on understanding interactions between training sets and downstream eval predictability. For example, training on RedPajama allows for predicting relative error for the 7B run on ARC-Easy at $\sim 5\%$; however, the prediction error is much higher for C4 and RefinedWeb trained models at $>26\%$. While influence functions [1, 2] are one promising avenue; we believe there is much more work to be done here.
>
> **Out-of-distribution (OOD) scaling clarification.** We provide the key takeaways for our OOD scaling study in L432-436, repeated below for convenience:
>
> >To probe the limits of reliable scaling, we attempt to break our scaling laws in out-of-distribution settings. We find that models trained on C4---English filtered---and evaluated on next token prediction on code domains have a high relative error in many cases. Perhaps surprisingly, evaluating the same models on German next token prediction gives reliable loss scaling (Figure 10).
>
> For more results see L1602-1615.
>
> -----
> **Additional references.**
> [1] Pang Wei Koh and Percy Liang. Understanding Black-box Predictions via Influence Functions. 2017.
> [2] Roger Grosse et al. Studying Large Language Model Generalization with Influence Functions. 2023.

---

> > ### Comment · Reviewer_fbs9 · 2024-11-26
> > **Re: Official Comment by Authors**
> >
> > Thank you for your clarification.

---

### Official Review · Reviewer_5cNe · 2024-11-04

**Soundness:** 4
**Presentation:** 3
**Contribution:** 3
**Rating:** 6
**Confidence:** 5

**Summary:**

This paper highlights two issues overlooked in current discussions of scaling laws for large language models: (1) To reduce inference costs, smaller models are often trained on more data, and (2) scaling law research commonly uses perplexity to measure model performance, whereas real-world evaluations focus on task completion performance, such as accuracy. In response, this paper investigates and confirms scaling laws in large language models related to over-training and performance on downstream tasks.

**Strengths:**

- The motivation is clear and the investigation is significant. Exploring scaling laws that are closer to real-world application scenarios provides strong guidance for the experimental setup of pre-training large language models.
- The experimental logic is very clear. The authors first selected certain experimental settings to fit and obtain hyperparameters for the specific scaling law. Then, they validated its accuracy on over-training and downstream performance across additional experimental settings, which is a reasonable approach.

**Weaknesses:**

- Some experiments are missing. The authors’ initial motivation for studying the scaling law in over-training was to reduce inference costs, but this raises a question: training smaller models on more data achieves performance comparable to larger models under the same compute budget. However, this issue is not well explained—are there relevant experiments or clear prior studies addressing this?
- The experimental validation of the scaling law for downstream performance is not rigorous enough. As shown in Table 2, there is a significant gap between the predicted and actual values of downstream performance based on the scaling law for individual datasets, while the average gap across 17 tasks is relatively smaller. Does this indicate that predictions from the downstream performance scaling law carry substantial risk? More specifically, we do not know for which specific number or types of tasks this scaling law provides accurate predictions.

**Questions:**

- Line 69-70: "M = N/D" seems to be a typo. It should be "M = D/N."
- Is Figure 1 a log-log plot?

---

> ### Author Response · Authors · 2024-11-25
>
> **How does over-training reduce inference cost?** Thanks for the chance to clarify! The fact that over-training reduces inference cost is something that can be shown by way of example. Consider a Chinchilla optimal model $A$ that has $N$ parameters and is trained for $D$ tokens. This model takes approximately $6 \cdot N \cdot D$ FLOPs of compute to train. Now consider a second model $B$ that has only $N/c$ parameters, where $c>1$, but that is trained on $k \cdot c \cdot D$ tokens, where also $k>1$. This model takes $6 \cdot k \cdot N \cdot D > 6 \cdot N \cdot D$ FLOPs to train (i.e., it is more expensive to train). Consider that $B$ is over-trained s.t. it matches $A$ in validation loss (i.e., performance). Model $B$ is *cheaper* to serve at inference, as it has fewer parameters and hence requires less inference FLOPs to run the forward pass. Looking at Figure 2 and considering a slice of the y-axis, we notice that this phenomenon does happen in practice.
>
> **When should we trust downstream scaling laws?** Thanks for the question! As pointed out by Schaeffer et al. (2023) specific downstream metrics can be hard to predict due to the non-linearity of top-1 error/accuracy metrics (L469). Our main finding is that the *average* error can be predictable. We still caution against using our downstream scaling laws for individual tasks. We do note; however, that average or aggregate metrics are very useful in practice to judge quality. For example, Li et al. [1] consider average metrics to guide their development of better datasets for language model training.
>
> **Is Figure 1 a log-log plot?** Figure 1 (left) is log-log, while Figure 1 (right) is linear. We hope that the axis scaling from matplotlib makes this clear; however, please let us know if there is something we can do to improve clarity here, and we are happy to apply the change!
>
> **Typo.** Thanks for pointing out the typo, we have fixed it in the updated manuscript.
>
> -----
>
> [1] Li et al. DataComp-LM: In search of the next generation of training sets for language models. 2024.

---

> > ### Comment · Reviewer_5cNe · 2024-12-03
> >
> > Thank you very much for your detailed response. I will raise the overall rating.

---

### Author Response · Authors · 2024-11-25

We thank the reviewers for their attention to our work, constructive comments, and positive feedback. Specifically, we are grateful for all reviewers highlighting our empirical efforts (5cNe, fbs9, tXac, HT5J). We are also appreciative of their mentioning the relevance of our scaling study in practical scenarios (5cNe, fbs9) and strong presentation (5cNe, fbs9, HT5J).

We also value the reviewers pointing out room for improvement. In the comments below we address their raised concerns.

Thanks again to all reviewers for their effort during the review process!

---

### Meta-Review · Area_Chair_nEbT · 2024-12-09

**Metareview:**

This paper fills a gap in the current scaling laws literature related to overtraining of smaller models. It measures downstream task performance, rather than simply aggregated perplexity.

Pros: Comprehensive experiments on a topic of practical interest to the machine learning community.

Cons: The individual task performance seems to be much less predictable than aggregated performance, and might not be described by scaling laws. In this sense, the paper appears to be somewhat overclaiming by claiming scaling is reliable.

There are also some strong assumptions involved in theory, specifically that some coefficients match.

**Additional Comments On Reviewer Discussion:**

After one reviewer brought up post training as a complicating factor, the authors argued that post training is cheaper than pretraining and therefore not a consideration. In practice, however, post training frequently involves human intervention and feedback, which is obviously very expensive in the data regime.

One reviewer raised their score in response to some clarifications, and all reviewers are in consensus that the paper is worth accepting.

---

### Decision · Program_Chairs · 2025-01-22

Accept (Poster)